# Droplet-based screening of phosphate transfer catalysis reveals how epistasis shapes MAP kinase interactions with substrates

Remkes A. Scheele[1,4], Laurens H. Lindenburg[1,4], Maya Petek [1,2,4], Markus Schober[1], Kevin N. Dalby [3] & Florian Hollfelder [1✉]

The combination of ultrahigh-throughput screening and sequencing informs on function and intragenic epistasis within combinatorial protein mutant libraries. Establishing a droplet-based, in vitro compartmentalised approach for robust expression and screening of protein kinase cascades (>10[7] variants/day) allowed us to dissect the intrinsic molecular features of the MKK-ERK signalling pathway, without interference from endogenous cellular components. In a six-residue combinatorial library of the MKK1 docking domain, we identified 29,563 sequence permutations that allow MKK1 to efficiently phosphorylate and activate its downstream target kinase ERK2. A flexibly placed hydrophobic sequence motif emerges which is defined by higher order epistatic interactions between six residues, suggesting synergy that enables high connectivity in the sequence landscape. Through positive epistasis, MKK1 maintains function during mutagenesis, establishing the importance of co-dependent residues in mammalian protein kinase-substrate interactions, and creating a scenario for the evolution of diverse human signalling networks.

[1] Department of Biochemistry, University of Cambridge, Cambridge CB2 1GA, UK. [2] Faculty of Medicine, University of Maribor, SI-2000 Maribor, Slovenia. [3] Division of Chemical Biology and Medicinal Chemistry, The University of Texas at Austin, Austin, TX 78712, USA. [4] These authors contributed equally: Remkes A. Scheele, Laurens H. Lindenburg, Maya Petek. ✉email: fh111@cam.ac.uk

Cell signalling coordinates cellular actions in response to external stimuli. Extracellular signals are transduced from membrane receptors to transcription factors via complex signalling networks, typically through protein kinase-mediated phosphorylation cascades. These coupled enzymatic networks have evolved to govern numerous critical cellular processes from cell proliferation to apoptosis[1]. Understanding the fine-tuning of complex signalling networks is a challenge that has mainly been addressed by reductionist biochemical approaches, i.e., studying individual protein mutants and/or single components of these networks at a time[2,3]. High-throughput screening and deep-sequencing analysis hold the potential to analyse a wider range of mutations in combination. Such insight is becoming increasingly important for the consideration of intragenic epistasis, i.e., the nonadditive interactions of amino acids within a protein[4].

This work addresses the interplay between activity and sequence in a human protein kinase cascade, investigating interactions between protein kinases by monitoring the downstream phosphorylation product of the cascade. So far, human protein kinase libraries have been screened with throughputs of ~$10^3$ variants, focusing on single-site saturation libraries[5–8]. Capturing epistasis, however, requires the screening of combinatorially randomised libraries, in which the exponential number of mutants with each added position quickly necessitates a higher experimental throughput. In vivo screening of four randomised interface positions in a bacterial histidine kinase (>$10^5$ variants) has enabled untangling of epistasis between said residues, although cellular background from acetyl phosphate was shown to phosphorylate target proteins as well, complicating the screening for kinase activity[9,10]. Insight into higher-order epistasis in human protein kinases would therefore benefit both from an increase in throughput, and ideally be derived from a cell-free system to avoid interference from the cellular machinery.

Here we establish an in vitro evolution system for protein kinase cascades with ultra-high throughput (>$10^7$). The screen is used to gain insights into the determinants of protein–protein interactions that control the catalytic activity of a mitogen-activated protein kinase (MAPK) cascade and the underlying epistatic effects between six residues that govern it.

Several mitogen-activated protein kinase (MAPK) pathways operate side by side in the cell[11,12] (Fig. 1a). To prevent inappropriate cross-talk between signalling responses, mitogen-activated protein kinase kinases (MKKs) contain a docking domain (D-domain) which targets a D-domain recruitment site (DRS) in MAPKs, facilitating binding and allosteric activating of their cognate partners[13,14]. (Fig. 1b[15]). Here, we randomise the docking domain of MKK1 (>500,000 variants), and test activation of the downstream kinase ERK2 by detecting its phosphorylation product in vitro. Selecting active mutants followed by deep sequencing gives us comprehensive insight into which combinations of residues in the MKK1 D-domain promote efficient phosphate transfer between MKK1 and ERK2, their robustness to mutation and the sequence landscape describing this interaction.

## Results

**Design of an in vitro screening system for a synthetic MAPK cascade.** We established a screening system for in vitro testing of a model protein kinase cascade (Fig. 1c) by immobilising genes encoding the chosen kinase (MKK1) on paramagnetic beads[16], together with a phosphorylation detection mechanism (subGFP, see below) (Supplementary Fig. 1A). The beads were co-compartmentalised in droplets with a purified downstream kinase (ERK2) and in vitro transcription and translation components (IVTT) (Supplementary Fig. S1B), so that polydisperse droplets contain at most one bead each (Supplementary Fig. 2).

Robust in vitro expression of the upstream kinase (MKK1) begins the signalling cascade, such that MKK1 phosphorylates the hydroxyl residues T185 and Y187 of ERK2[17]. Activated ERK2 could now phosphorylate an N-terminally immobilised proline-directed substrate peptide (VA**PFSP**GGRAK)[18], labelled with a C-terminal GFP (subGFP).

At a chosen timepoint, the emulsion was broken to give a mixture of beads, which were now differentially phosphorylated, representative of each MKK1 variant's activity and complementarity to ERK2. To detect phosphorylation of the subGFP's serine, the beads were incubated with chymotrypsin: it specifically cleaves the substrate peptide after Phe as long as the peptide is not phosphorylated. In contrast, ERK2's phosphorylation of serine sterically/electrostatically hinders chymotrypsin cleavage, leaving the peptide intact and GFP still attached to the bead[18]. Ultrahigh-throughput sorting of library members (each represented by one bead carrying genotype and phenotype) can then be performed by flow cytometry, where the beads encoding the best catalysts show the highest fluorescence and mutants are separated into 'bins' according to their activity. Finally, sequence recovery and next-generation sequencing of the encoded kinase variant couples the genotype to the activity-defined bin.

**Assaying activity and binding complementarity in in vitro MKK1-ERK2 signalling.** The activation of the investigated phosphorylation cascade depends on the binding complementarity between the two kinases and on the enzymatic activity of the associated individual components. To probe whether this in vitro, bead-based screening approach can select for either aspect, first individual emulsions were prepared with beads encoding MKK1 variants with different constitutive activities, IVTT mix, and ERK2 (Fig. 2a). The beads retained subGFP fluorescence intensity in accordance with the activity of the encoded variant: (i) beads encoding the constitutively active variant $^{ca}$MKK1 (i.e., the mutant MKK1$^{\Delta44-51/S218D/M219D/N221D/S222D}$)[19] retained ~50% subGFP fluorescence intensity, (ii) beads encoding a constitutively active variant with approximately eightfold lower relative activity for ERK2 (MKK1$^{S218D/S222D}$)[19] retained ~16%, and (iii) the wild-type MKK1 without constitutive activation retained <1% of subGFP fluorescence intensity relative to that seen with untreated beads. Likewise, the activity of the downstream cascade component is critical: when it is compromised by incubating the highly active $^{ca}$MKK1 with the phosphorylation-defective variant ERK2$^{T185A/Y187A}$ [20], the percentage of retained subGFP drops to <1%.

Next, utilising the most active $^{ca}$MKK1 variant, we probed the in vitro screen's ability to enrich for $^{ca}$MKK1 variants based on their docking interactions with ERK2. To do so, we first prepared a binding-impaired variant by altering the $^{ca}$MKK1's D-domain to contain two alanine residues (Ile9Ala and Leu11Ala, hereafter denoted $^{ca}$MKK1$^{I9A/L11A}$), impairing its ability to target the DRS of ERK2. These mutations have been shown to reduce the relative activity of MKK1 for ERK2 > 2-fold[21] (Fig. 2b, all residues numbered according to MKK1). Beads were functionalised with either $^{ca}MKK1$-Cy5 or $^{ca}MKK1^{I9A/L11A}$-TexasRed (TxR) genes (Supplementary Fig. 3a–d) and the beads mixed in a 1:1 ratio before emulsification. After de-emulsification and chymotrypsin digest, flow cytometric analysis allows separation of $^{ca}MKK1$-Cy5 and $^{ca}MKK1^{I9A/L11A}$-TxR based on their fluorescent dye, before assessing the subGFP intensities of the beads, indicative of efficient phosphate transfer in the cascade (see the Gating strategy in Supplementary Fig. 3a–f).

The impaired D-domain of $^{ca}$MKK1$^{I9A/L11A}$ slows activation of ERK2 relative to $^{ca}$MKK1, an effect best visualised by the bimodal distribution of subGFP-negative and subGFP-positive beads shifting over time (Supplementary Fig. 3g–k). The bimodal

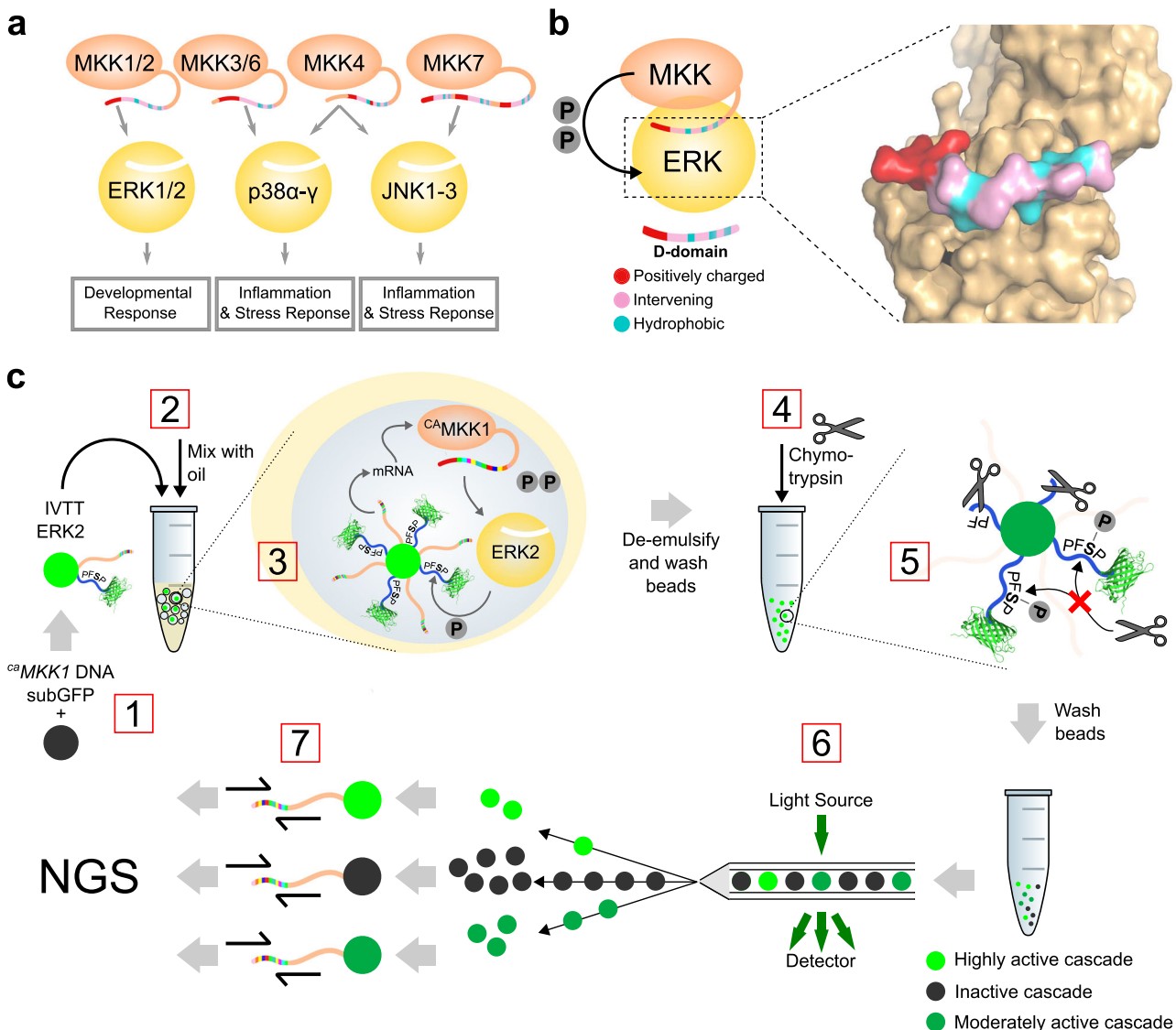

**Fig. 1 Overview of Human MKK–MAPK interactions and screening approach for MKK1-ERK2. a** A schematic overview of cognate MKK interactions with MAPKs and their downstream effectors. The D-domain is visualised as the multi-colour section of the unstructured N-terminal region of MKKs. **b** Phosphorylation of ERK1/2 by MKK1/2 is facilitated by MKK's D-domain binding the D-domain recruitment site (DRS) of ERK2. Shown here is the D-domain sequence of MKK2, the residues highlighted in red bind an acidic patch of residues in the ERK2 DRS, while the large hydrophobic residues highlighted in aqua bind hydrophobic grooves within the ERK2 DRS. Spacer residues are shown in pink (PDB: 4H3Q)[15]. **c** (1) Paramagnetic beads (green circle) are functionalised with subGFP (blue wavy line with GFP) and $^{ca}$MKK1 genes (pink wavy lines). (2) Beads are emulsified by mixing an aqueous solution containing in vitro expression components (IVTT) and purified ERK2 protein with oil and surfactant. (3) Expression of $^{ca}$MKK1 in the emulsion droplets starts the cascade by phosphorylation of ERK2. Phosphorylated ERK2 is now able to phosphorylate serine within the substrate sequence of the subGFP construct. (4) After de-emulsification, beads are exposed to chymotrypsin protease (denoted as scissors) in bulk. (5) Beads with phosphorylated serine will be less susceptible to digestion by chymotrypsin, leaving subGFP attached to the bead. The bead, therefore, encompasses both the genotype and phenotype information, as an encoded active $^{ca}$MKK1 mutant will retain more subGFP on a bead than a less-active kinase mutant. (6) Subsequent flow cytometric sorting of the beads based on subGFP fluorescence into activity-set gates followed by (7) deep next-generation sequencing (NGS) allows for correlation of the cascade activity to the encoded MKK1 gene.

distribution is likely to stem from the exponential activation of subGFP via activated ERK2, as the substrate (ERK2) is an enzyme, and will continue to phosphorylate subGFP upon activation by MKK1. At the optimal timepoint ($t = 3$ h, including time for MKK1 expression in droplets), the sequence-dependent D-domain function was robustly resolved (Fig. 2c, d), where subGFP-positive beads had a 92 ± 2% positive predictive value for encoding $^{ca}$MKK1 over $^{ca}$MKK1$^{I9A/L11A}$, which indicates a 8% false positive rate (Fig. 2e, raw data in Supplementary Fig. 4c–i). The high positive predictive value enables reliable enrichment for functional D-domains when sorting subGFP-positive beads

downstream. At the same time, good coverage when screening a D-domain library can be achieved by as little as three-fold oversampling (Supplementary Fig. 4; coverage >95% of variants, limited by the percentage of false negatives in Q1: 35 ± 27%).

**Combinatorial D-domain library design and MKK1-ERK2 interaction screening.** The D-domain sequence comprises a set of basic, spacer and hydrophobic residues (Fig. 1b). In vivo, both MKK1 and MKK2 phosphorylate ERK2. Compared to MKK1, the MKK2 D-domain is two residues longer, containing an additional leucine at position 7a and an alanine at 8a (Fig. 2b). To

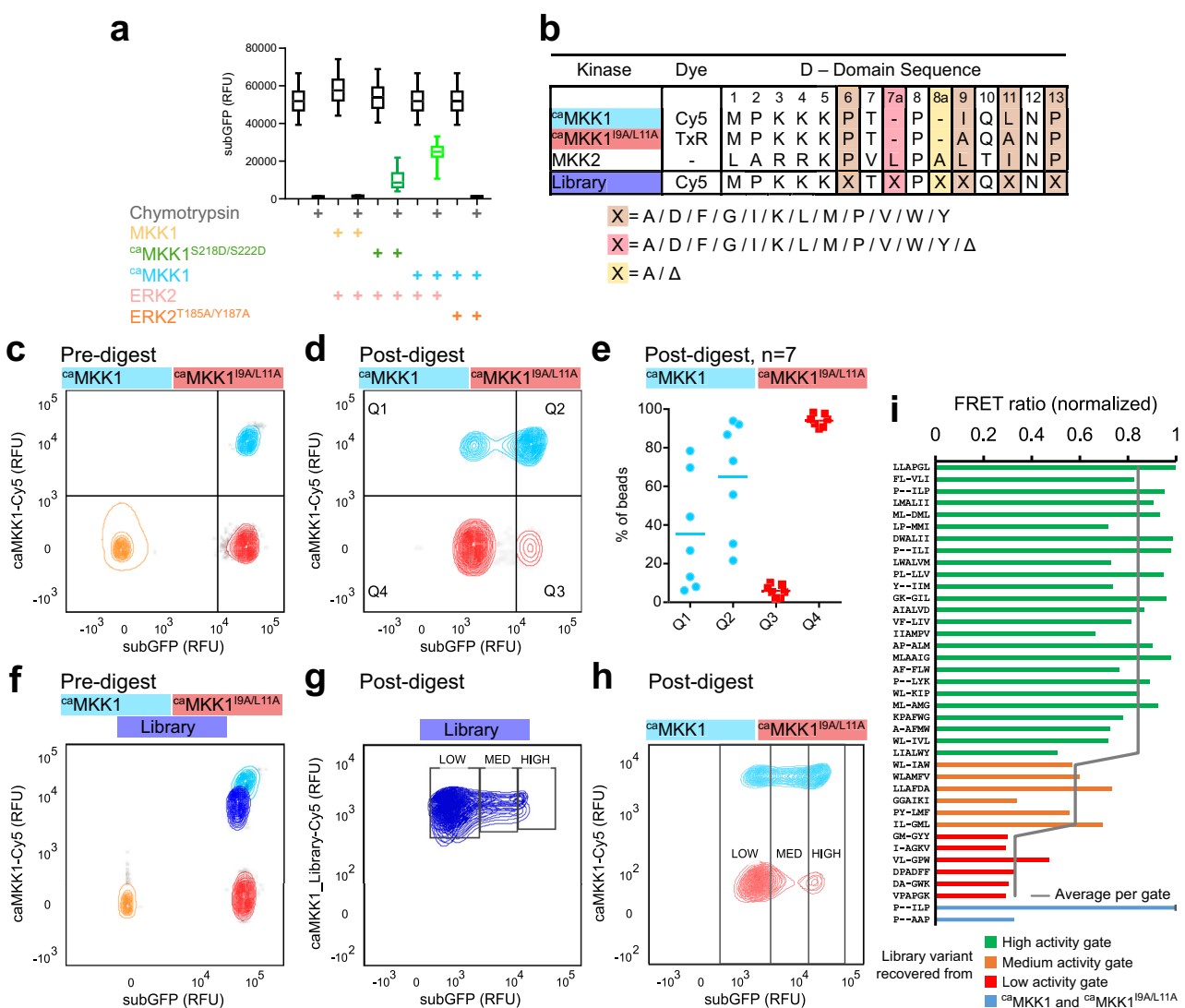

**Fig. 2 Assay of reconstituted synthetic cascades in IVTT emulsion droplets. a** GFP retention of beads encoding non-constitutive active (*MKK1*) or constitutively active (*caMKK1*/*caMKK1 S218D/S222D*) genes in combination with ERK2/ERK2 T185A/Y187A. Box and whisker plot shows 5th, 25th, 75th, 95th percentiles and the median for n = ~10,000 screened beads per sample. **b** D-domain sequences and design of the randomised D-domain library of caMKK1. Numbering is based on MKK1's WT sequence, 'a' indicates a potential insertion, while 'Δ' indicates a "deletion" of this residue when the insertion was not introduced. **c** Flow cytometric analysis of a 1:1 mixture of beads functionalised either with active caMKK1 (aqua) or with docking domain impaired caMKK1 I9A/L11A (red) before and (**d**) after chymotrypsin digest. Non-functionalised beads are shown in orange. Analytical gates around the bimodal distribution distinguish between false negatives (Q1), true positives (Q2), false positives (Q3) and true negatives (Q4) (**e**) The percentage of beads for caMKK1 and caMKK1 I9A/L11A in the analytical gates Q1–Q4 over seven independent measurements, of which one is shown in 'D' and all raw data is shown in Supplementary Fig. S4. The average percentage is denoted as a horizontal line. **f** Analysis of SpliMLiB library preparation by flow cytometry (dark blue), where positions have been randomised as described in panel b. The controls, beads functionalised directly with the full-length genes via a one-step click-reaction are shown in aqua (caMKK1, positive) and red (caMKK1 I9A/L11A, negative). Non-functionalised beads are shown in orange. **g** An overlay of the nine library sorts (Supplementary Fig. S7) shown in blue and the respective gates in which the low/medium and high GFP beads were sorted (**h**) An overlay of the three caMKK1 (aqua) vs caMKK1 I9A/L11A control samples (Supplementary Fig. S7) prepared in parallel to the library samples. **i** FRET sensor assay for randomly recovered caMKK1 sequences from the high-activity (green), medium activity (orange) and low activity sorting gate (red). All values are normalised to caMKK1. Averages derived from two biological repeats are shown. Only randomised residues are shown in sequence labels. Source data are provided as a Source Data file for panels **a**, **e**, and **i**.

investigate the importance of both the D-domain's length and chemical composition for ERK2 interaction, we created a library of the caMKK1 docking domain. The library contained four independently randomised hydrophobic/spacer residues, while additionally allowing for the incorporation of two insertions (MKK2-like), one insertion or none (MKK1-like), for a total of six randomised residues (Fig. 2b). In order to stay within bounds of the screening capacity, a subset of amino acids was screened at

each position. Positions were chosen to ensure the inclusion of amino acids with polar, charged, or hydrophobic residues, with a wider range of possible hydrophobic substitutions as they complement the hydrophobic pockets in the ERK2 DRS (Fig. 1b). The gene was assembled using SpliMLiB, an unbiased combinatorial gene assembly method, on paramagnetic microbeads[22]. SpliMLiB results in a dense monoclonal coating of DNA on each bead, ensuring the efficient expression of a unique MKK1 variant from

each bead by IVTT. The capture of subGFP and library synthesis was shown to be successful through flow cytometry (Fig. 2f and Supplementary Fig. 5).

The starting library contained 500,000 variants and was assayed in nine parallel batches. We recovered 3.5 million beads after de-emulsification and chymotrypsin digest, yielding seven-fold oversampling (Supplementary Fig. 6). These beads were sorted with flow cytometry into three activity bins; low, medium and high $^{ca}$MKK1 activity, measured through subGFP fluorescence (Fig. 2g, individual plots in Supplementary Fig. 7). The low-, medium- and high-activity gates in which beads were to be sorted were set relative to the bimodal distribution of three independent bead populations encoding either $^{ca}$MKK1 or $^{ca}$MKK1$^{I9A/L11A}$, assayed in parallel with the library samples (Fig. 2h, individual plots in Supplementary Fig. 7). Based on the control distributions observed for $^{ca}$MKK1 and $^{ca}$MKK1$^{I9A/L11A}$ encoding beads, we expect the false positive rate of the high gate to be ~9%, which matched our selective pressure predictions described above. Our false negative rate of finding $^{ca}$MKK1-functionalised beads in the low or medium gate was estimated to be ~50%, which, with sevenfold oversampling, means we expected >99% of variants with functioning D-domains to be sorted at least once in the high gate for subsequent sequencing of its genotype. The majority of library beads were sorted into the low activity gate ($2.1 \times 10^6$ or 85%), $2.8 \times 10^5$ (11.2%) were sorted into the medium activity gate, and $8.1 \times 10^4$ (3.2%) into the high-activity gate.

We further validated the ultra-high-throughput kinase screen by expressing and individually re-screening a random subset of sorted $^{ca}$MKK1 variants from each gate in a plate-based secondary screen. In the secondary screen, the in vitro-expressed $^{ca}$MKK1 was mixed with ERK2 and assayed against the same target peptide used in the ultra-high-throughput screen, here placed between CFP and YFP in a FRET sensor[18]. The efficiency of phosphate transfer is read out by monitoring the emission ratio after complete chymotrypsin digest: while the un-phosphorylated sensor is readily digested and gives a low-emission ratio, the sensor phosphorylated by an active kinase is digested more slowly (Supplementary Fig. 8). The 25 randomly selected variants from the high-activity gate retained on average 84% of the emission ratio signal normalised to $^{ca}$MKK1, confirming that our sort enriched for $^{ca}$MKK1 variants with wild-type-like activity (Fig. 2i). In contrast, the medium gate random variants retained on average 58% of the FRET signal, while only two of the low gate variants showed a slight improvement compared to the FRET assay when done with $^{ca}$MKK1$^{I9A/L11A}$. Thus we were confident that the in vitro assay and sort enriched for stable, catalytically activating protein–protein interactions.

**Deep sequencing of $^{ca}$MKK1 variants evaluates sequence preferences**. Next, we recovered the D-domain coding sequence from each activity bin independently through limit-cycle PCR (Supplementary Figs. 9 and 10), and analysed the sequence space of MKK1-ERK2 docking complementarity through deep sequencing of the sorted variants from all three gates (Supplementary Tables 1–3). Accounting for overlap between the sequencing pools, we detected $4.9 \times 10^5$ unique sequences in total or 91% of the theoretical library diversity (Fig. 3a). The SpliMLiB protocol generates balanced high-quality libraries with equal proportions of randomised amino acids at every targeted position[22]. Sequence analysis of the low activity gate distribution, a proxy for an unselected library, shows the expected balance: the low activity sample contained the bulk of the library ($4.7 \times 10^5$ sequences, or 95% of the total number of unique sequences recovered, including the expected false negatives (Fig. 3a) and

exhibited a balanced distribution of amino acids (Shannon entropy >0.995 at all positions, close to the value of 1.000 that would indicate a perfectly balanced library)[23] (Supplementary Figs. 11 and 12). The amino acid proportions at each randomised position showed increasing enrichment towards preferred residues in the medium and especially the high-activity gate (Shannon entropy 0.86–0.99), giving a clear view on which amino acids confer activity and tend to be enriched at each site in the D-domain of MKK1 (Supplementary Fig. 11).

As both highly active and medium activity variants may be observed at low frequency in the high gate (Supplementary Figs. 13 and 14), we next focused on 29,563 variants that showed a similar or better sequencing count profile as WT $^{ca}$MKK1. This set of active variants contains variants that appear ≥51 times in the high gate and are enriched in the high gate over the lower gates (see Supplementary Note 1 for details). The conservative approach to variant filtering removed 19% of the sequences in the high gate, slightly above the expected 8% false positive sorting events (see "Methods" and Supplementary Fig. 13). Alternative formulations of the enrichment ratio, based on the distribution of sequencing reads in all gates, and/or dividing over all observed sequences, yielded near identical results (Supplementary Note 2 and Supplementary Fig. 15). Overall, the D-domain sequences in the final active dataset ($2.9 \times 10^4$ variants) show strong enrichment of Leu and Ile at randomised positions 9 and 11 (Fig. 3b). This trend follows the wild-type residues in MKK1/MKK2 and conforms with the highly conserved nature of these residues[3,24], and their complementarity to the binding pockets on ERK2 (Fig. 1b). Similarly, we observed a preference for the MKK2-like hydrophobic residue insertions at position 7a, favouring especially the insertion of Leu. This third Leu in MKK2's D-domain (L7a) is complementary to an additional hydrophobic pocket in the ERK2-binding groove[2] (the "lower pocket"; Fig. 1b). Thus, the large hydrophobic residues at MKK1 positions 7a, 9 and 11 form the core of the D-domain features that target the ERK2 DRS.

The randomised positions at the edge of the randomised region show mild overall sequence preferences: whereas Pro is present in positions 6 and 13 in both the wild-type sequences of MKK1 and MKK2, the deep mutation scan reveals a weak preference for Pro at position 6 and a general enrichment for any hydrophobic residue at position 13. As expected, the negatively charged residue Asp was depleted across the library, especially in the core hydrophobic patch positions 7a, 9 and 11. Positively charged Lys was neutral in position 6, next to the already basic patch of amino acids in MKK1, and depleted otherwise. Similar to charged residues, the very small residues glycine and alanine were depleted in all positions, consistent with the known effect of alanine to disrupt MKK1 binding, as shown in the negative control construct $^{ca}$MKK1$^{I9A/L11A}$ (Fig. 3b).

**Positive epistasis widens D-domain active sequence space**. The sequences in the final set of active variants contained 13 variants with a single-site mutation relative to wild-type MKK1 (Supplementary Fig. 16), while 43 out of the remaining 44 possible MKK1 single mutants were abundant in lower activity gates (Supplementary Fig. 16). Similar to the overall preference for Leu/Ile residues in active D-domains, the point substitutions in these 13 variants tend to be hydrophobic. If only these substitutions were allowed in a combinatorial fashion, we would expect to observe at most 432 ($=2 \times 3 \times 1 \times 3 \times 4 \times 6$) unique active variants in the final active deep-sequencing dataset: [6: IP][7a: KLΔ][8a: Δ][9: FIP][11: FIML][13: FILPVW]. However, the number of active variants in the final active set of active variants was an order of magnitude larger ($2.9 \times 10^4$ variants), which indicates the

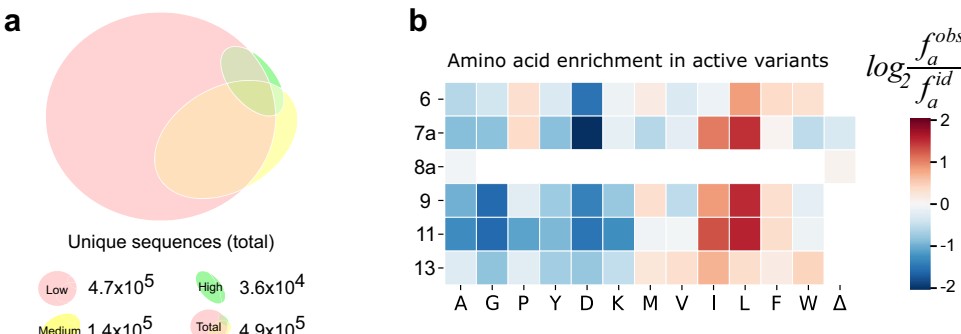

**Fig. 3 The amino acid distribution observed in deep sequencing of the activity-sorted MKK1 variants. a** Proportional Venn diagram displaying the number of unique sequences found in the low activity gate (≥3 sequencing reads, red), medium activity gate (≥5 sequencing reads, yellow) and the high-activity gate (≥10 sequencing reads, green). **b** The enrichment of observed amino acids ($f_a^{obs}$) at each position in the final active set of n = 29,563 variants found in the high gate, relative to expected frequencies ($f_a^{id}$) in a perfectly balanced library.

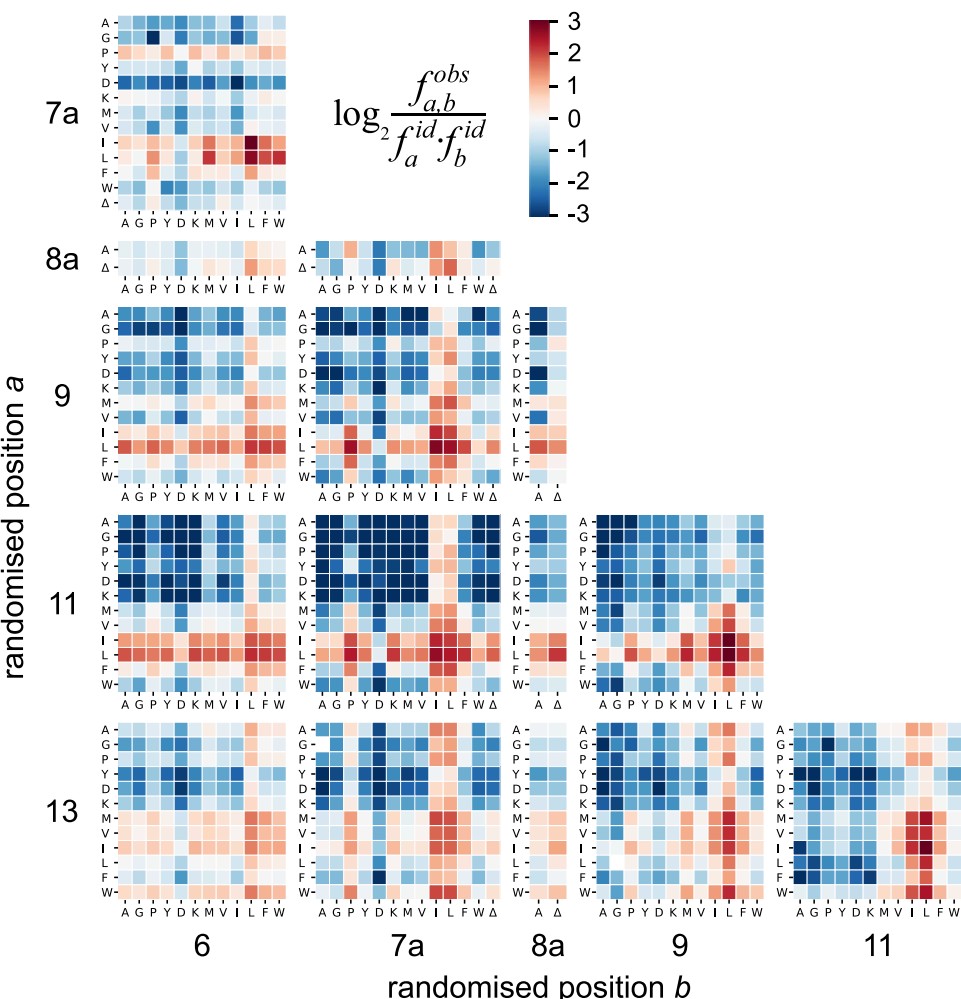

**Fig. 4 Pairwise enrichment.** The heatmaps show how the observed frequency of amino acid residues in two randomised positions ($f_{a,b}^{obs}$) divided by the expected frequency in an ideally balanced library ($f_a^{id} \cdot f_b^{id}$). A logarithmic scale is used to display both enrichment (red) and depletion (blue). Each panel represents one pair of randomised positions. The presence of large hydrophobic residues Leu/Ile at any one of positions 7a, 9 or 11 serves as an "anchor", so that these residues appear in combination with non-preferred amino acids. n = 29,563 for each individual heatmap.

presence of positive epistatic interactions, where pairs of mutations exhibit a more beneficial effect than the sum of their individual contributions in isolation[25].

While hotspot analysis of the $2.9 \times 10^4$ variants already paints a more accurate estimation of the general amino acid preferences at each of the randomised positions (Fig. 3b) since it encompasses

non-linear effects, it does not provide insight into which particular pairwise interactions are epistatically intertwined. To draw out the underlying epistasis from this one-dimensional model we mapped the prevalence of all pairs of mutations in the D-domain (Fig. 4 and Supplementary Fig. 17). We observed that for some pairs of MKK1 positions, the preferred residues at one

position are influenced by the amino acid present at the other. While most position pairs followed the trends typical for each randomised position (i.e., enrichment of hydrophobic residues), we observed clusters of enrichment wherever a Leu/Ile is present, indicating that previously determined non-beneficial residues such as Ala are allowed in the context of a hydrophobic anchor residue. Further quantitative investigation of shifts in amino acid preference supports the interpretation that the system exhibits widespread epistasis and that the two-position enrichments are not just a reflection of the hydrophobic residue abundance, as the Bonferroni-adjusted p-values for the $\chi 2$ goodness of fit comparison are $<10^{-15}$ for all pairs of positions (Supplementary Figs. 17, 18, Supplementary Table 8 and Supplementary Note 3). This prevalence of positive epistasis indicates that the presence of an Ile or Leu residue allows anchoring to ERK2 in combination with otherwise deleterious residues, such as alanine or aspartate. Interestingly, these co-enrichment patterns are also present for Leu in position 6 and all hydrophobic residues in position 13, positions with a weak overall sequence preference. This ability of the hydrophobic amino acid residues to expand the sequence space further supports the hypothesis that these residues are essential to D-domain binding and activation. Conversely, the preferred large hydrophobic residues show weak negative epistasis with each other, suggesting that a D-domain full of hydrophobic residues is not necessarily better.

**Preferred motifs in the D-domain.** Having shown that the amino acid preferences at the six randomised positions are interdependent and epistatically linked, we searched for structural motifs within the set of 29,563 variants that underpin this pattern. We first examined the change in preferences when one or more residues are restricted to preferred hydrophobic residues, comparing the change in amino acid distribution with the overall sequence preferences (Supplementary Fig. 19). If a Leu or Ile is fixed in any one position, there is a shift in the sequence preference in the adjacent randomised position towards hydrophobic residues, creating an Φ–X–Φ motif (Φ denoting Leu or Ile, and X the fixed spacer residue). Conversely, the sequence preference at distant positions is relaxed and becomes more tolerant of small or hydrophilic residues.

While a single large hydrophobic residue is not sufficient to anchor a D-domain, two such residues arranged in an Φ–X–Φ motif greatly reduce constraints on the rest of the D-domain sequence (Fig. 5a). Conversely, if two adjacent positions contain other residues, the remaining randomised positions are strongly enriched for Leu and Ile. The effect of the Φ–X–Φ motif is strongest in positions 7a/9, 9/11 and 11/13, where D-domains typically contain hydrophobic residues and the short Φ–X–Φ motif with two Leu/Ile residue appears sufficient. The activating effect of two hydrophobic residues is apparent regardless of the exact position of the motif, although the Φ–X–Φ motifs in positions 6/7a and 11/13 tend to be supplemented by a third hydrophobic residue nearby (Supplementary Fig. 20). The incorporation of two small or charged residues in positions 7a and 11 would prohibit the formation of any Φ–X–Φ motif, which explains the strong negative epistasis between non-preferred residues in these two positions (Supplementary Fig. 17).

Normalising the subsets of Φ–X–Φ containing datasets for the initial distribution of amino acid frequencies (Fig. 3b) gave us a glimpse into multidimensional epistasis i.e., the positive or negative magnitude epistasis with a third residue outside of the Φ–X–Φ motif (Fig. 5b). The emergence of the distinct Φ–X–Φ motif showcases how multidimensional epistasis shapes the sequence preference for a distinct Φ–X–Φ motif in active MKK1 D-domain sequences. Whenever a Φ–X–Φ is present

(may it be at positions 6-X-7a, 7a-X-9, 9-X-11 or 11-X-13) negative epistasis is observed with a third hydrophobic at any other position, while positive epistasis is observed with any third residue which is non-beneficial in isolation. Likewise, when residues other than Ile/Leu are present at these positions, positive epistasis is observed with Ile and Leu at all other positions in the subset, meaning that the frequency of finding a Ile/Leu goes beyond the already abundant frequencies of Ile and Leu at these positions in the entire active variant dataset. On the contrary, non-beneficial amino acids become ever more detrimental in this context through negative epistasis.

Thus, the D-domain sequence landscape is shaped by the requirement for a two-residue hydrophobic motif, although the location of this motif in the D-domain is flexible, an effect which was not immediately apparent from hotspot analysis (relying on additivity). Accommodating this shifting motif is thought to rely on the promiscuity of the ERK2 DRS, which naturally accommodates several unique D-domains with different orientations to maximise overlap with its hydrophobic pockets[2]. The co-evolved amino acids which complete the motif are less constrained in chemical functionality but likely optimise packing to the DRS—indicated by the strong negative epistasis of multiple small residues co-occurring (Supplementary Fig. 17).

**Sequence similarity network shows evolutionary connectivity.** Having observed that an active D-domain can accommodate the key hydrophobic Φ–X–Φ motif in more than one possible position, we addressed the clustering of all 29,563 active variants. We constructed a sequence similarity network (SSN) and visualised it with a force-directed layout (Fig. 6). Each variant in the final active dataset is a node and edges connect all pairs of variants that differ by a single-site mutation. We were interested in whether the SSN contains any distinct regions or clusters within the network, which would indicate the possibility of distinct biological solutions to the problem of ERK2 activation. Therefore, the largest connected subgraph of the full SSN—spanning 97% of nodes in the full SSN—was partitioned using the Leiden algorithm to identify clusters within the network[26]. Here, a cluster is a collection of nodes (protein variants) that are highly connected to each other but have relatively few links to nodes in other clusters.

This SSN contains nine clusters with substantial membership, with a modularity score of 0.75 indicating that the clusters are well-defined (possible values between −0.5 and +1, where a higher value indicates stronger divisions into clusters). The clusters are further grouped into two equally sized groups separated by the presence or absence of the 8aAla insertion (Fig. 6). Thus, the 8aAla insertion appears to divide the sequence space, likely through forcing a different conformation in the DRS-binding pocket[27]. Within each region of the sequence landscape, the clusters are distinguished by the nature and position of the defining single large hydrophobic residue. The Φ–X–Φ motif is apparent in 7 out of 9 clusters, typically with one residue strictly defining the cluster (e.g., 11 L in yellow and green clusters) and the second residue one of several choices. Two clusters show a slightly different motif: the cyan variants share the 9L-X-11F motif and the smallest, black cluster is defined by the 9M-X-11I anchoring motif—in each case, one I/L residue is substituted for a chemically similar large hydrophobic residue (Fig. 6).

Examination of the network structure shows that this SSN is not scale-free; that is, there are no clear hub variants in this network[28]. Instead, the D-domain SSN predominately features numerous moderately well-connected variants (Supplementary Figs. 21 and 22), which could be explained by the number of dimensions that are simultaneously probed. When screening multiple site saturations in parallel, fitness peaks might disappear

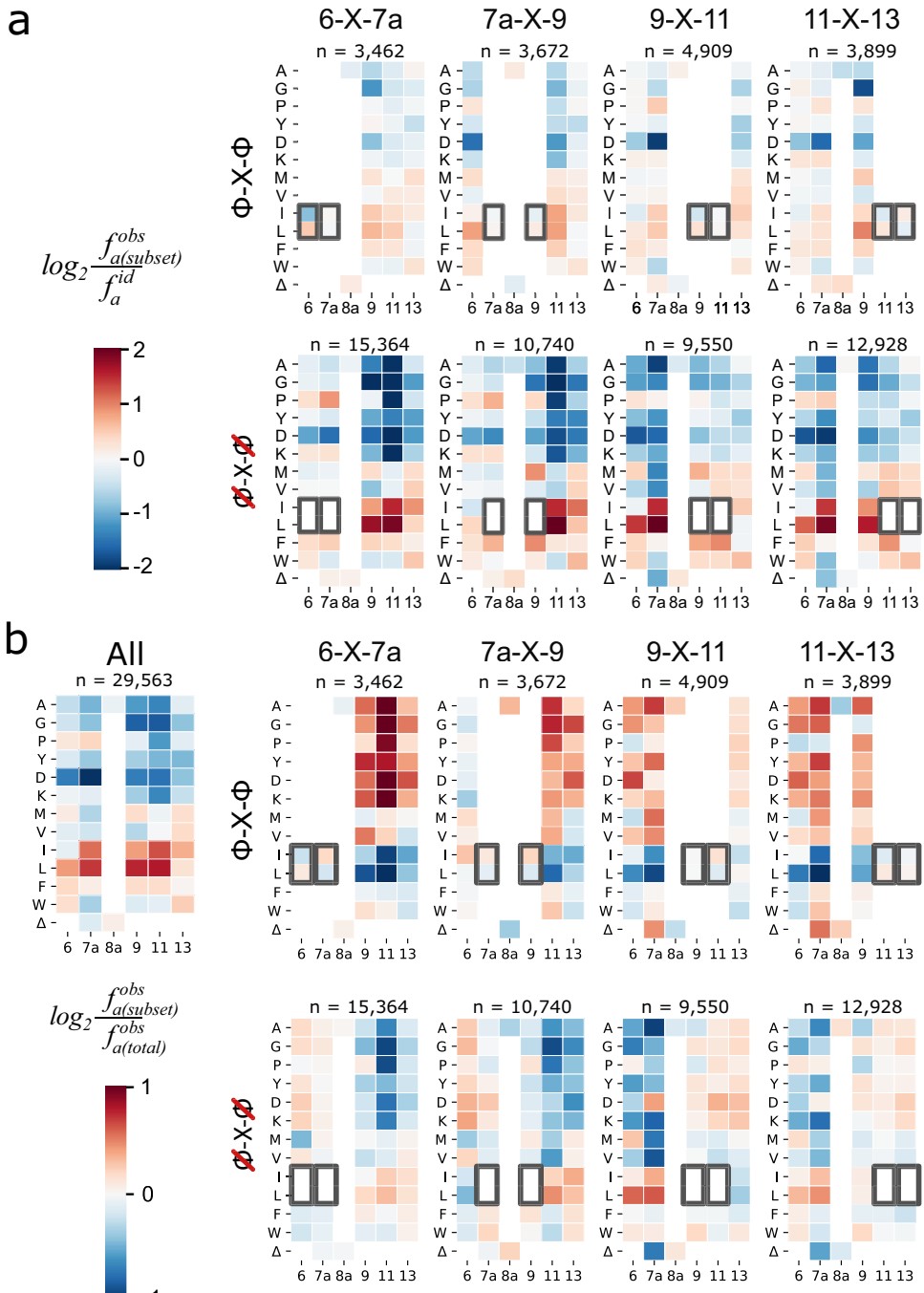

**Fig. 5 Multidimensional epistasis shapes the formation of a Φ–X–Φ motif in the D-domain. a** Heatmaps are shown for subsets of all 29,563 active sequences, which either include an isoleucine/leucine (Φ) at positions 6 and 7a, 7a and 9, 9 and 11, or 11 and 13 (top row), or the inverse, including anything but isoleucine/leucine (Φ̸) at those positions. Subsets that contain a Φ–X–Φ motif have washed out colours in the other positions indicating that the sequence preference the other positions is relatively weak. When anything but Φ is present in these subsets, strong enrichment for Φ is observed at the other randomised positions. **b** The heatmap in the top left shows the enrichment for all 29,563 variants. All other heatmaps are epistasis plots, which like (**a**), are comprised of subsets of the 29,563 active sequences. Here, the amino acid frequencies in the chosen subset of variant (containing/excluding the Φ–X–Φ motif) are divided by the amino acid frequencies in the full set of active variants; consequently, the heatmaps show the change in preference compared to the full dataset. The top row shows that when the Φ–X–Φ residues are included, there is an epistatic preference for non-hydrophobic residues (Ala, Gly, Pro, Tyr, Asp or Lys) in the rest of the D-domain. Conversely, if we examine the variants without the Φ–X–Φ motif on one end, there is an additional preference for the motif elsewhere in the domain, beyond the general enrichment for Leu/Ile residues.

as a result of neutral ridges that connect these fitness peaks in the multidimensional fitness space[29,30]. As a result, Gavrilets's "holey landscape" model predicts high fitness genotypes (in this case, the 29,563 variants) to permeate all regions of genotype space,

reducing the expected number of clusters seen in a SSN. Here, the many solutions for binding and activation of human kinases established by positive epistasis are well connected, allowing kinases to roam genotype space more freely, potentially aiding the

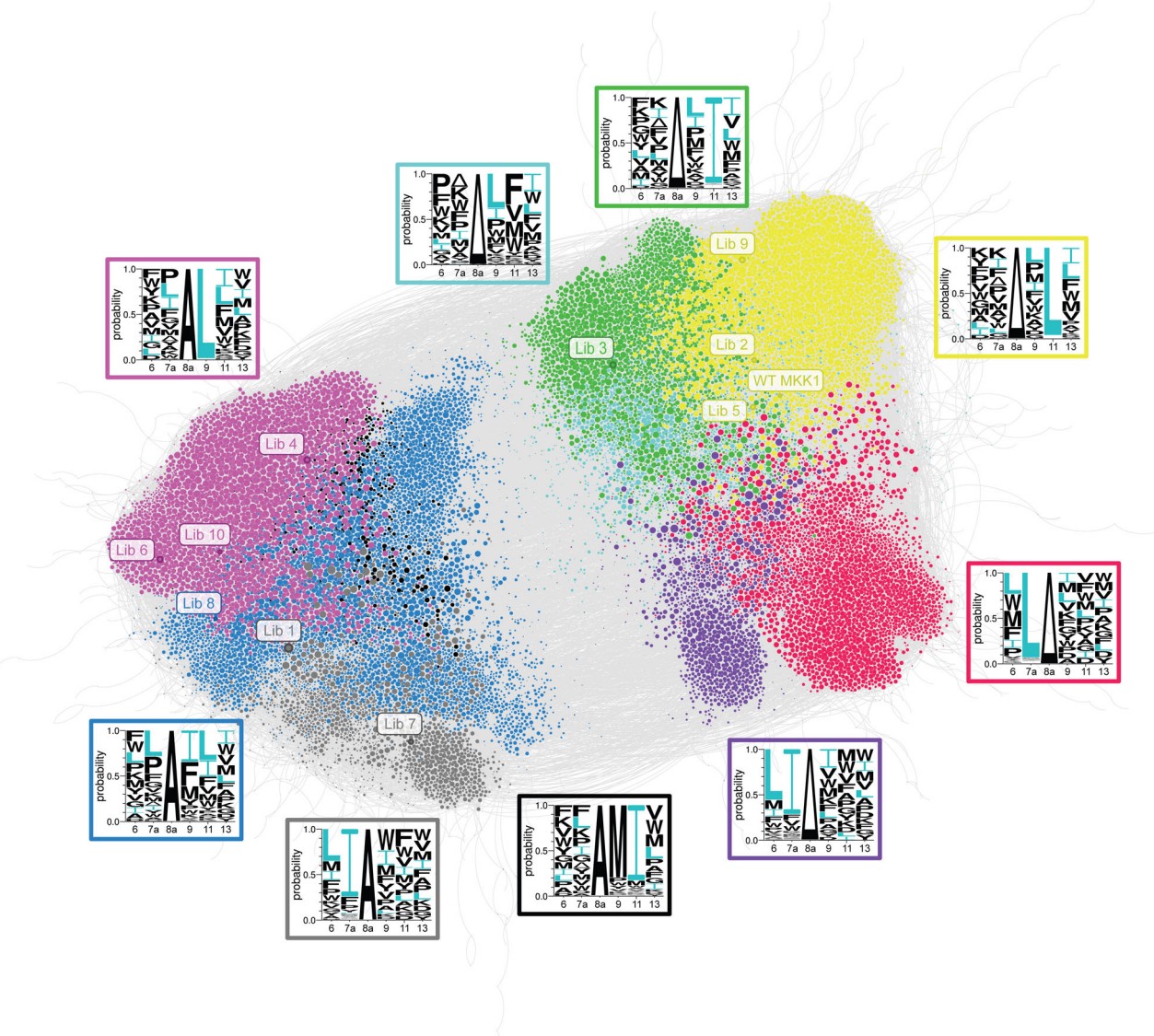

**Fig. 6 The key hydrophobic residues appear in different positions in the sequence similarity network (SSN).** The SSN is constructed with each active variant as a node and an edge connecting every pair of variants that differ at exactly one position. Partitioning the SSN with the Leiden algorithm detected nine substantially populated clusters in this SSN, such that the variants are coloured according to cluster membership and node size is proportional to the number of active variants accessible from that variant via a single mutation. The location of the library variants used for affinity screening is indicated. The Ala insertion at position 8a causes a registry shift which partitions the SSN in two equal-sized parts. Within each half, each cluster is distinguished by a sequence fingerprint (WebLogos in boxes) with different anchoring hydrophobic residues.

process of diversification or evolution of novel signalling pathways.

**Analysis of synthetic D-domains from the sequence network.** After examining the general features of the D-domain sequence landscape, we investigated the affinities of D-domains towards the ERK2-binding groove from different regions in the sequence space (Fig. 6). The affinities of the synthetic D-domain peptides were assayed in a previously established fluorescence anisotropy-based assay, where non-labelled D-domains compete for binding to the ERK2 DRS with a fluorescent D-domain (fl-HePTP) specific for the ERK2 DRS[2] (Supplementary Figs. 23 and 24). Similar to the wild-type MKK1 D-domain, all 10 D-domains selected from the library out-competed fl-HePTP from the binding groove, whereas the D-domain with I9A and L11A mutations did not. Interestingly, the variant most enriched in the final active dataset bound ERK2 with an affinity higher than the WT MKK1

D-domain (Lib1, Supplementary Fig. 24). In line with the ubiquitous epistasis observed in the MKK1 D-domain, this is not merely dependent on additivity of Ile/Leu (when multiple Leu or Ile residues are present): a consensus D-domain containing the most enriched residue at each position (Leu-D, Supplementary Fig. 24) had a lower binding affinity for the ERK2 DRS than the WT MKK1 D-domain or any other library selected D-domain.

## Discussion

The development of a in vitro ultra-high throughput assay, in tandem with a highly randomised combinatorial library design, enabled us to perform a comprehensive analysis of the epistatic sequence determinants of binding and catalysis in the MKK1 docking domain. As an in vitro screen for reaction turnover for eukaryotic kinases that reaches such throughput (~$10^7$ variants per day, with minimal reagent consumption—less than 200 µL total), this assay makes biological phosphate transfer practically

amenable not only to sequence exploration (as in this study) but also in the future to directed evolution. This work demonstrates the screening for the MKK1-ERK2 cascade, but testing other kinase cascade elements could be extended with a suitable cognate target peptide downstream, when designed to be susceptible to protease cleavage.

The in vitro character of screening in droplet compartments is experimentally liberating: screening campaigns that are otherwise hindered by difficult heterologous expression[31,32] or functional redundancy and conflicting kinase activity from background kinases in the expression host[9,10,33], will become possible in a cell-free environment[34]. Their readout should report directly on intrinsic properties of the proteins probed, i.e., protein–protein interactions, catalysis and regulation.

Such intrinsic effects have emerged in this study, defining allowed motifs and evolvability by characterising sequence diversity in six randomised positions: active docking D-domain sequences are strongly enriched for the $\Phi$-X-$\Phi$ motif, which can be placed anywhere along our tested sequence space and predicts productive phosphate transfer between two kinases in the MAPK pathway. It is possible that this insight might have emerged in a lower throughput study, albeit with less statistical significance. However, in addition to identifying a sequence feature, the deep-sequencing data illuminate a comprehensive, six-dimensional combinatorial sequence landscape: single Leu/Ile residues in isolation and the $\Phi$–X–$\Phi$ motif in particular re-shape the rest of the D-domain preference through widespread positive epistasis.

Previous sequence-function mapping studies have largely scanned the effect of single-site mutations across full genes[35]. Datasets that include an analysis of epistatic effects have mostly focused on pairwise interactions only[25,36], such as in the determination of structural interactions[37,38], While higher-order epistasis (i.e., interactions between more than two residues) has been recognised as important in adaptive evolution and key to determining evolutionary trajectories[36,39], investigations have focused on small recombination libraries[25,40], while few have examined higher-order epistasis exhaustively with high-throughput experiments[9,41–47].

In the MKK1 docking domain, widespread positive epistasis generates evolutionary contingency, avoiding loss of function and enabling smooth transitions between residue combinations. Consequently, a type of sequence landscape emerges in which the functional variants are highly interconnected but do not contain any clear hubs, which could be explained by the Gavrilets Holey Landscape model where the presence of neutral ridges in multidimensional fitness space allows variants to permeate all regions of sequence space. In our data, mutational tolerance is epistatically enhanced in the presence of a Leu/Ile residue pattern and the $\Phi$–X–$\Phi$ motif that promotes adaptability through positive epistasis. These features may help explain how diverse signalling cascades emerged in evolution. In addition to the versatility that the recognised modularity of kinases provides[48], our observations suggest that epistatic effects contribute to robust intrinsic evolvability of protein function.

The experiments described in this work mimic a neutral drift scenario in evolution (a non-adaptive exploration of sequence space) that plays a prelude to future adaptive evolution by creating a multitude of starting points for functional innovation[49,50]. When the trade-off between mutational burden and retention of function in this neutral phase is favourable, successful adaptive evolution becomes more likely[51]. Such evolutionary contingency will smoothen evolutionary transitions to expand kinase cascades that govern cellular development, stress responses and maladaptive states such as cancer[52]. One of the aims of synthetic biology is the construction of synthetic networks (e.g., operating in engineered T-cell therapies) with complex and predictable outputs[53].

More generally, our approach exemplifies how the functionality of a sequence motif can be dissected, categorised and understood by carrying out essentially just one miniaturised ultra-high-throughput experiment followed by sequence interpretation. Ultra-high-throughput technologies are key in this endeavour: the >100-fold greater throughput compared to previous human kinase deep- sequencing experiments[5–8] has yielded additional insight and at the same time provided, compared to traditional experimentation, a richer dataset suitable for systematic examination of sequence landscape parameters.

## Methods

**Protein expression and purification**. All DNA and protein sequences are denoted in Supplementary Table 4. The MAPK *ERK2* construct (UNIPROT ID P28482) was cloned as part of a pET28a expression plasmid. Oligonucleotides used for site-directed mutagenesis to generate *ERK2*$^{T185A/Y187A}$ were designed with NEBaseChanger. The subGFP construct was cloned into a pHATA[54] expression vector resulting in pHAT-Avi-subGFP, where subGFP was in frame with an N-terminal His6 and Avi-tag. The FRET sensor for secondary validation was cloned into the expression vector pOP3BT, containing an N-terminal His8 tag.

Plasmids harbouring ERK2 or FRET sensor constructs were transformed into electrocompetent *E. coli* BL21(DE3) (NEB). pHAT-Avi-subGFP was transformed into *E. coli* BL21(DE3) that had been co-transformed with plasmid overexpressing BirA. All constructs were expressed in 500 mL LB with 50 µg/mL kanamycin (MAPKs), 100 µg/mL carbenicillin (FRET sensor), or 100 µg/mL carbenicillin and 25 µg/mL chloramphenicol (subGFP), induced at OD$_{600}$ 0.4–0.6 with 0.1 mM IPTG (+40 µg/mL biotin for subGFP) and grown overnight at 25 °C.

All proteins were purified using standard Ni affinity chromatography. ERK2 was dialysed to Phosphate Buffered Saline (154 mM NaCl, 5.60 mM Na$_2$HPO4, 1.06 mM KH$_2$PO$_4$ without calcium and magnesium, from here referred to as PBS (Lonza)) with 2 mM DTT for all experiments except for fluorescence anisotropy, for which ERK2 was dialysed to PBS_A (PBS, 2 mM DTT, 2 mM TCEP, 2 mM EDTA, 2 mM EGTA and 0.1% (v/v) Brij35). The FRET sensor was dialysed to 20 mM Tris-HCl (pH 8.0) 100 mM NaCl. subGFP was dialysed to PBS. All proteins were aliquoted for single-use, flash-frozen and stored at −80 °C.

**Preparation of full-length *MKK1*, $^{ca}$MKK1$^{S218D/S222D}$, $^{ca}$MKK1 and $^{ca}$MKK1$^{I9A/L11A}$ amplicons**. All DNA and protein sequences are shown in Supplementary Table 4. $^{ca}$MKK1 and *MKK1* genes (UNIPROT ID Q02750) were part of a pIVEX2.3d IVTT expression vector without purification tags. The $^{ca}$MKK1$^{I9A/L11A}$ and $^{ca}$MKK1$^{S218D/S222D}$ mutations were introduced via Agilent QuikChange in the $^{ca}$MKK1/MKK1 genes respectively. Full-length gene fragments for bead functionalisation were prepared using 800 µL PCR reactions consisting of 0.5 µM of each forward (LMB_Cy5/LMB_TexasRed) and dibenzocyclooctyl (DBCO)-linked reverse primer (T7T_DBCO) (Supplementary Table 5), 1x BIOTAQ NH4 buffer, 3 mM MgCl$_2$, 1 mM dNTPs, 0.5 ng/µL plasmid template and 0.05 units/µL BIO-TAQ DNA polymerase (BIOTAQ DNA polymerase and buffer were from Bioline, London, England). Thermocycling was performed starting with 2 min at 96 °C, followed by 30 cycles of 15 s at 96 °C, 15 s at 55 °C, 90 s at 72 °C, followed by a final extension step at 72 °C for 5 min.

The PCR fragment was purified using Solid Phase Reversible Immobilisation (SPRI), and eluted with MilliQ (0.02% (v/v) Tween-20)[22]. Typically, 800 µL of PCR product was eluted with 40 µL MiliQ (0.02% (v/v) Tween-20) to concentrations of ~1 µg/µL.

**Bead functionalisation with full-length *MKK1* amplicons and subGFP**. The paramagnetic carboxy-beads (Ø 5 µM; S1964, microParticle, Berlin) were first functionalised with azide and Tamavidin-2-HOT as previously described[22]. For click chemistry-mediated coupling of DBCO-DNA and azide, beads were washed in 1.5-mL Eppendorf tubes with 500 µL PBS (to which was added 0.05% (v/v) Tween-20), on a DynaMag-2 (ThermoFisher) magnetic rack which allowed separation of the magnetic beads from the buffer. This was followed by a washing step with 200 µL binding buffer (Dynabeads kilobaseBINDER kit).

The coupling reaction of the full-length, DBCO-linked *MKK1* amplicons to azide-modified beads used ~2 × 10$^7$ DNA fragments per bead for complete saturation, assuming the average molecular weight of each base pair to be 650 Daltons. For example, a 100 µL solution of $^{ca}$MKK1 amplicons (1660 bp, 1.5 µg/µL) would be used to functionalised 4 × 10$^6$ beads—adding up to 2.1 × 10$^7$ DNA fragments per bead.

Beads were mixed with equal volumes of Binding Solution (Dynabeads kilobaseBINDER Kit, ThermoFisher) and purified DNA. The beads were incubated in a shaker (90 min, 1150 rpm, 37 °C). The supernatant was removed using the DynaMag-2. The beads were washed with 40 µL of Washing Solution (Dynabeads

kilobaseBINDER kit) while immobilised to the magnet without resuspension. Next, the beads were washed as usual with PBS (0.05% v/v Tween-20).

Biotinylated subGFP was diluted in PBS (0.05% (v/v) Tween-20) (15 μM, 300 μL) was resuspended with the Tamavadin-2-HOT containing beads, and incubated in a shaker (30 min, 1150 rpm, 20 °C). After removing the supernatant, beads were washed 3× with PBS (0.05% (v/v) Tween-20).

**Preparation of SpliMLiB fragments.** SpliMLiB library assembly has been described previously[22]. SpliMLiB design and oligonucleotides used are shown in Supplementary Fig. 25 and Supplementary Table 6.

*Preparation of fragP13X and fragP6X.* All 12 P13X fragments were amplified in parallel, 800 μL PCR reactions consisting of 0.5 μM of each forward (P13X) and reverse (T7T_DBCO) primers, 1× BIOTAQ NH4 buffer, 3 mM MgCl2, 1 mM dNTPs, 0.5 ng/μL *ca*MKK1 pIVEX template and 0.05 units/μL BIOTAQ DNA polymerase (BIOTAQ polymerase and buffer were from Bioline, London, England). Thermocycling was performed starting with 2 min at 96 °C, followed by 30 cycles of 15 s at 96 °C, 15 s at 55 °C, 60 s (for fragP13X) or 20 s (for fragP6X) at 72 °C, followed by a final extension step at 72 °C for 5 min. The PCR fragment was purified using Solid Phase Reversible Immobilisation (SPRI), and eluted with MilliQ (0.02% Tween-20)[22]. Typically, 800 μL of PCR product was eluted with 40 μL MilliQ (0.02% Tween-20) to concentrations ~1 μg/μL.

*Preparation of fragL11x, frag(A8a)_I9X and frag(7aX).* All reverse primers (12 for fragL11x, 24 for frag(A8a)_I9X, 13 for frag(7aX) were phosphorylated. All forward primers followed the same protocol, except that no T4 PNK was added to the reaction. Reactions (30 μL, in MilliQ) consisted of 1× T4 ligase buffer (NEB) (3 μL from 10× stock), 15 μM oligo (4.5 μL from 100 μM stock) and 1.5 μL T4 PNK. For forward oligos, the T4 PNK was replaced with 1.5 μL MilliQ. All 98 samples were incubated for 1 h at 37 °C, and 20 min at 65 °C. Of each forward and reverse primer, 25 μL was combined and annealed by incubating the mixture for 2 min at 95 °C, 10 min at 52 °C, and ramping down to 4 °C. All annealed oligos were kept at 4 °C ready for ligation.

**SpliMLiB assembly.** The 12 unique fragP13X oligos were attached via click-mediated chemistry as described for the full-length *MKK1* amplicons.

*Digestion of beads.* Beads were washed with 1 × 3.1 buffer (NEB) in 0.02% v/v tween. To the beads, 360 μL digestion mixture was added, consisting of 12 μL BspQI, 1 × 3.1 buffer (NEB) in 0.02% (v/v) Tween-20. The beads with immobilised fragP13X were resuspended, divided in three PCR tubes, and incubated for 90 min at 50 °C. Afterwards, beads were washed with PBS (0.05% (v/v) Tween-20) and stored at 4 °C until further use.

*Ligation of fragL11X, frag(A8a)_I9X and frag(7aX).* The beads were divided over a number of tubes given by the number of fragments in the next round (12 for fragL11X, 24 for frag(A8a)_I9X and 13 for frag(7aX)). To each, 100 μL of ligation mixture was added consisting of 45 μL annealed oligonucleotide, Na-ATP (0.45 mM final), DTT (0.5 mM final), 10× T4 Ligase buffer (NEB) (0.5x final), 5 μL T4 Ligase (NEB) in 0.02% (v/v) Tween-20. The mixture was incubated for 30 min at 20 °C. Afterwards, the beads were washed with PBS (0.05% (v/v) Tween-20). The next round would start with another round of digestion of beads, after which the next set of fragments could be ligated.

*Digestion and ligation of the final fragment P6X.* After ligation of frag(7aX) and subsequent digestion of the beads, the last fragment could be ligated on. FragP6X was first digested in twelve parallel reactions consisting of the purified oligo (~20 μL of 400 ng/μL diluted in 40 μL 1 × 3.1 buffer in 0.02% v/v tween with 4 μL BspQI. The mixtures were incubated for 90 min at 50 °C, and purified using silica columns (Zymoclean Gel DNA Recovery, Zymo Research). Zymo PCR clean-up kit, and eluted in 10 μL MilliQ to give ~200 ng/μL solutions of the digested P6X fragments. The digested beads which contained the sticky overhang from the previously ligated fragment, frag(7aX) were divided over 12 PCR tubes. To this, 100 μL ligation mixture was added, consisting of 10 μL (~2 μg) of the digested, purified fragP6X, in 1× T4 in 0.02% (v/v) Tween-20 with 5 μL T4 Ligase. The beads were washed with PBS (0.05% (v/v) Tween-20), pooled and stored at 4 °C. Finally, subGFP was attached, as described for beads functionalised with full-length *MKK1* amplicons above.

**Emulsification of beads with IVTT and ERK2.** Beads (0.5–1 × 10^6) containing *MKK1* DNA (prepared either as full-length amplicons or through SpliMLiB), and subGFP were resuspended in 12.5 μL PUREExpress (the IVTT mix); 5 μL solution A, 3.75 μL solution B, 0.5 μL murine RNAse inhibitor (15 U, NEB), 1 μL 18.75 μM ERK2 (1.5 μM final), and 2.25 μL nuclease-free water. Quickly, RAN surfactant (RAN biotechnologies) (1%) in HFE 7500 (Fluorochem) (100 μL) was added to the mixture. The emulsion was formed by repeated aspiration and expiration of the oil and aqueous phase through a 50 μM filter contained in a custom made 200 μL pipette tip until the mixture was homogenous[16]. The emulsion was kept on ice

(10 min) until the oil and emulsion layers had separated, after which the emulsions were incubated in a thermocycler at 30 °C until de-emulsification.

**De-emulsification and chymotrypsin digest.** Before de-emulsification, the excess oil below the emulsion layer was removed using a Gel-Saver-Tip II (200 μL). PBS (0.05% (v/v) Tween-20) (100 μL) and 1H,1H,2H,2H-Perfluorooctanol (PFO) (Alfa Aesar) PFO (18 μL) were added to the emulsion and mixed through manual rotation of the tube. After phase separation, the aqueous layer containing the beads was carefully removed to avoid oil contamination. This process was repeated three times.

The beads were washed in chymotrypsin cleavage buffer (0.5 M Tris-HCl (pH 8.0), 5 mM CaCl, 0.05% (v/v) Tween-20) and digested in the same buffer with 40 nM chymotrypsin (22 °C, 1100 rpm, 30 min). Finally, the beads were washed in PBS (0.05% (v/v) Tween-20), ready for flow cytometry.

**Flow cytometric analysis and FACS.** Flow cytometric analysis was carried out on a FACScan Cytek machine for beads stored in PBS (0.05% (v/v) Tween-20). SubGFP fluorescence was quantified using 488 nM excitation, with a 525/40 bandpass filter. TexasRed fluorescence was quantified using 561 nm excitation, with a 610/20 bandpass filter. Cy5 fluorescence was quantified using 640 nm excitation, with 660/10 bandpass filter. Flow cytometric sorting of beads was performed on a BD FACSAria Fusion, with four-way sorting into different tubes according to subGFP fluorescent intensity. SubGFP fluorescence was quantified using 488 nM excitation, with 530/30 bandpass filter. Cy5 fluorescence was quantified using 640 nm excitation, with 660/20 bandpass filter. The forward and side scatter profiles were used to ensure sorting was restricted to single beads, while Cy5 gating was used to select for beads that contained full-length *ca*MKK1 library DNA.

**Recovery of full-length *MKK1* amplicons from sorted beads for the secondary FRET assay.** The beads sorted into the high, medium and low activity gates were pooled accordingly and washed with PBS (0.05% (v/v) Tween-20). From the pooled beads, ~5% (v/v, after vortexing) were transferred for full-length recovery by PCR, whereas the other 95% (v/v) was used for recovery of the D-domain sequence (see below). The bead populations were washed with 1x Q5 buffer (NEB). The full-length *MKK1* genes were recovered using Q5 High-Fidelity 2x Master Mix (NEB), using the primers FL_Rec_F and FL_Rec_R (Supplementary Table 5). The PCR tubes containing the beads in 1× Q5 buffer were placed on a DynaMag-96 Side Magnet (ThermoFisher), and the supernatant was carefully discarded. To each tube, a 100 μL reaction mastermix was added consisting of 0.5 μM of each forward and reverse primer and 1× Q5 MM (NEB). Thermocycling was performed starting with 30 s at 98 °C, followed by 30 cycles of 10 s at 98 °C, 15 s at 67 °C, 60 s at 72 °C, followed by a final extension step at 72 °C for 2 min. The PCR mixture was purified over silica columns (Clean & Concentrate, Zymo Research). The PCR recovery product was gel extracted and purified using silica columns (Zymoclean Gel DNA Recovery, Zymo Research). The PCR fragment was cloned by Gibson assembly[55] into an empty, linearised pET28a expression vector, amplified with the primers pET28_Rec_F and pET28_Rec_R (Supplementary Table 5). The plasmid population was transformed to α-Select Chemically Competent E. coli (Bioline, now discontinued). Colonies were picked to inoculate a 5 mL LB_kan culture. The next day, the plasmid was purified using Miniprep (Qiagen), and sequenced. Plasmids that contained *MKK1* genes with mutations outside of the D-domain (an artefact of amplification by BIOTAQ polymerase, see above) or not containing the *MKK1* insert were discarded from further analysis.

**Secondary FRET assay.** The unique MKK1 variants, recovered from the respective gates, were expressed using PureExpress (NEB) according to the manufacturer's protocol (incubating for 2 h at 37 °C). In total, 5 μL of the IVTT reaction were added to a 96-well plate and combined with ERK2 (4 μM), ATP (2 μM) (Na-salt, Sigma Aldrich), FRET construct (20 μM) and DTT (2 μM) for a plasmid reaction volume of 100 μL in 1× kinase buffer (50 mM HEPES (pH 7.5), 10 mM MgCl2, 1 mM EGTA, 0.01% Brij35, 0.01% (v/v) Tween-20). The plates were incubated for 30 min (at 30 °C), after which 50 μL chymotrypsin (30 nM final) was added in 2× chymotrypsin cleavage buffer (0.66× chymotrypsin buffer final). The loss of FRET ratio was measured over 30 min using a Tecan Infinite 200 Pro plate reader using a 420 nm excitation wavelength, calculating the emission ratio as defined in Eq. (1).

$$emission\ ratio = \frac{emission\ \lambda_{527nm}(RFU)}{emission\ \lambda_{475nm}(RFU)} \quad (1)$$

Cerulean (excitation maximum $\lambda_{433}$, emission maximum $\lambda_{475}$) transfers energy and excites Citrine (excitation maximum $\lambda_{516}$, emission maximum $\lambda_{529}$) so that a high emission ratio corresponds to the intact (phosphorylated) FRET sensor, and a lower emission ratio to the cut (non-phosphorylated) FRET sensor after exposure to chymotrypsin. The bandwidth of excitation/emission was 9/29 nm, respectively, with 25 flashes per measurement.

**Recovery of D-domain sequences from sorted beads and deep sequencing.** The number of PCR cycles for recovery of the D-domain was first optimised to prevent PCR-induced biases[56] by monitoring the reaction using quantitative

real-time PCR (qPCR) with EvaGreen dye (Biotium) from different template concentrations (numbers of beads containing full-length $^{ca}MKK1$ DNA) (Supplementary Fig. 9). A pre-exponential fluorescent threshold was set, and the number of cycles required to reach said threshold was correlated to the number of beads. The resulting formula ($r^2$ 0.99) was used to calculate the number of cycles required to reach the same threshold when recovering the D-domain from sorted library beads (Eq. (2)). In which $y$ equals the number of cycles and $x$ the number of beads.

$$y = -2.5 \ln(x) + 39 \qquad (2)$$

The pooled low ($2.1 \times 10^6$ beads), medium ($2.8 \times 10^5$ beads) and high bead population ($8.1 \times 10^4$ beads) were washed with 1× Q5 buffer, dividing each sample into eight reactions for recovery PCR. After removing the 1x Q5 buffer, a 100 μL reaction mastermix (consisting of 0.5 μM of each forward (NGS_F) and reverse primer (NGS_R) in a 1× NEBNext Ultra II Q5 Master Mix (NEB) was added to the beads (Supplementary Table 5). Thermocycling was performed starting with 30 s at 98 °C, followed by a variable number of cycles (8/13/16 for low/medium/high gate sorted beads, respectively) of 15 s at 98 °C, 15 s at 65 °C, 7 s at 65 °C, ending with a final extension step at 65 °C for 10 s.

The PCR mixtures were purified over silica columns (Clean & Concentrate, Zymo Research), pooling the eight individual PCR products for each gate. The amplified D-domains were multiplexed and processed into Illumina TruSeq libraries by the University of Cambridge, Department of Biochemistry Sequencing Facility according to the manufacturer's instructions. The libraries were sequenced with two Illumina MiSeq 2 × 75 bp runs (20% PhiX spike-in), which gave 10.5/4.2/6.2 million reads each for the low/medium/high gates.

**Deep-sequencing analysis.** The raw forward and reverse reads were merged using PEAR[57] and filtered for reads containing the correct 5- or 6-base forward or reverse sequence. Due to the close spacing of substitutions and insertions in a short sequence, the reads could not be sensibly aligned to reference as DNA sequence. Therefore, all following steps were performed on the protein sequences using a set of custom Python scripts, which are freely available at www.github.com/fhlab/Kinases.

The D-domain sequence was extracted from FASTQ files (using script "proteins.py") by searching all DNA reads with a regular expression for DNA sequence that matches the library design, and all such sequences were translated into protein sequence. The sequences were aligned to the reference sequence MPKKKXTPXQXNXAPDG using the Emboss implementation of Needleman-Wunsch algorithm[58] with a custom scoring matrix (Supplementary Table 7). Each amino acid match is scored equally positive (+5), all amino acid mismatches are penalised equally (−1) and a match to a randomised X position is scored as weakly positive (+1). Insertions and deletions are scored with gap open penalty 3 and gap extension penalty 5. This scoring scheme allows successful alignment of short, highly randomised sequences and correctly places the inserted residues (7a, 8a if present) after the randomised substitution position. The aligned sequences were parsed to extract and count all present mutations and stored as a dictionary for later analysis.

All three sequencing fractions were analysed to determine the number of observed variants, setting a cut-off of ≥ 10 sequencing reads for the high gate to reduce spurious observation of sequencing noise. However, because the medium and low gates are substantially more diverse, a lower cut-off of ≥ 3 supporting reads was used there. The amino acid diversity at each randomised position (Supplementary Fig. 11) was calculated by extracting the amino acid position at each randomised position and weighing the composition by the number of times each variant was observed in the fraction of interest.

**Statistics.** The variants observed in the high gate were filtered according to the number of reads in the high gate (51 or more) to generate a final active dataset of active variants. This threshold was chosen after consideration of multiple filtering schemes (see Supplementary Notes 1 and 2) to reflect the prevalence of the WT sequence in the high gate.

The single position enrichment was derived by dividing the observed proportion (frequency) of each possible amino acid, $f_a^{obs}$ (ranging between 0 and 1), by the ideal proportion $f_a^{id}$ that is expected in a balanced library ($f_a^{id} = \frac{1}{12}, \frac{1}{13}, \frac{1}{2}$ for positions 6, 9, 11, 13; 7a; and 8a, respectively). Similarly, the two-position enrichment was calculated by dividing the joint observed frequency of amino acids a and b, $f_{a,b}^{obs}$, by the ideal joint proportion $f_{a,b}^{id} = f_a^{id} f_b^{id}$ (in the ideal case, the probabilities are independent) (Eq. (3)).

$$enr_{joint} = \frac{f_{a,b}^{obs}}{f_{a,b}^{id}} \qquad (3)$$

The presence of epistatic interactions between positions (i.e., deviation from the independence of probabilities) is obtained by dividing the $f_{a,b}^{obs}$ ($= f_{a|b}^{id} f_b^{id}$) by the joint proportions expected from the one-dimensional enrichment at each position, given by $f_{a,b}^{obs} = f_a^{obs} f_b^{obs}$ without epistasis (Eq. 4) (see Supplementary Note 3 for detailed derivation). We repeated the analysis using the amino acid distribution in all observed variants instead of the ideal distribution and reached identical conclusions.

$$enr_{joint, epistatic} = \frac{f_{a,b}^{obs}}{f_a^{obs} \cdot f_b^{obs}} \qquad (4)$$

**Exploration of the sequence similarity network.** A sequence similarity network was constructed with Python using the NetworkX package (https://networkx.org/, version 2.4) and the open-source visualisation software Gephi (https://gephi.org/, version 0.9.2). Each variant in the final active set of active variants was added as a node and the Hamming distance between all pairs of variants was calculated; an edge was added to the network if the Hamming distance was equal to 1. The degree, betweenness, closeness and eigenvector network centrality scores were calculated, but they did not reveal any clear hub nodes.

We addressed the question whether network can be partitioned in a meaningful way in order to explore node connectivity by the Louvain method[59] for community detection, as implemented by the "community louvain" module (https://python-louvain.readthedocs.io/en/latest/). This shows that a standard Louvain partitioning had a good modularity score of 0.7 (range −0.5 to +1.0, a positive value is better). However, the Louvain algorithm only guarantees that each node is well connected to its own community, yet some communities might be poorly connected to themselves (that is, missing a bridge *within* the community). Such mis-assignment can happen, if the bridge node within a community is also well connected to nodes outside of the community: so that it gets reassigned to the outside community, leaving the original community divided. Therefore, we used the improved Leiden algorithm (method described by ref. [26], implemented in https://leidenalg.readthedocs.io/en/stable/intro.html), which identified nine large communities within the largest connected subgraph of the full network. The D-domain sequences in each community were extracted in FASTA format and used to create a WebLogo representation of sequence content[60].

**Peptide affinity assays.** The procedure has been adapted from Garai et al[2]. All synthetic peptides were ordered from GenScript with purities >95%. The fluorescent reporter peptide fl-HePTP was purchased with an N-terminal fluorescent dye (5-carboxyfluorescein). Fl-HePTP was dissolved in MilliQ water. All other D-domains were dissolved in PBS_A.

Reaction volumes for all experiments were 25 μL, prepared in a 384-well assay plate (Corning, type 3820). For $K_D$ determination of fl-HePTP, ERK2 in PBS_A was concentrated to 240 μM using Amicon Ultra—4 centrifugation filters (Amicon, 30 K). Protein was sequentially diluted in PBS_A, and mixed 1:1 with fluorescent reporter peptide. For $K_i$ determination of non-labelled D-domains, D-domains (dissolved to 1 mM) were serially diluted in PBS_A. A solution containing ERK2 (25 μM) and fl-HePTP (40 nM) (final concentrations) were mixed 1:1 with the D-domain dilutions. Corning plates containing the reaction mixtures were spun down (2 min, 4000 rpm). Fluorescence polarisation was measured using a PheraStar FS (BMG Labtech), with a FP 485 520 520 filters for 5-FAM HePTP.

**Reporting summary.** Further information on research design is available in the Nature Research Reporting Summary linked to this article.

## Data availability

The NGS data are available at the European Nucleotide Archive under accession number PRJEB39786. The processed NGS data are included with the source code (see below) in the "data" directory. Source data are provided with this paper.

## Code availability

The code is freely available at www.github.com/fhlab/Kinases (https://doi.org/10.5281/zenodo.5733133).

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

## Acknowledgements

We thank Esther Perez of the Cambridge NIHR BRC Cell Phenotyping Hub, the late Nigel Miller and Joana Cerveira of the Department of Pathology for help with flow cytometric analysis. This work was funded by the Horizon 2020 programme of the European Commission. L.L. and M.S. received individual postdoctoral EU Marie-Curie fellowships, R.A.S. was supported by a studentship from the EU ITN MMBIO and F.H. is an ERC Advanced Investigator (695669). M.P. was supported by a studentship [BB/M011194/1] from the Biotechnology and Biological Sciences Research Council. K.N.D. acknowledges support from the Welch Foundation (F-1390).

## Author contributions

Conceptualisation: L.L., M.S., K.N.D., F.H. and R.A.S.; methodology: L.L. and M.S. and R.A.S., M.P. and F.H.; investigation: R.A.S.; software: M.P.; data curation: M.P. and R.A.S.; writing—original draft: R.A.S.; writing—review & editing: R.A.S., M.P., L.L., F.H.,

K.N.D. and M.S.; visualisation: R.A.S. and M.P.; supervision: L.L., F.H. and K.N.D.; funding acquisition: F.H. and L.L.

## Competing interests

The authors declare no competing interests.
