## [Peer Review File · Nature Communications]

Droplet-based screening of phosphate transfer catalysis reveals how epistasis shapes MAP kinase interactions with substratesReviewers' Comments:

Reviewer #1:

Remarks to the Author:

The current studies present the development of a high throughput system to evaluate how intragenic mutations can regulate the activation of the extracellular signal-regulated kinase-2 (ERK2) by its upstream activator MKK1. This approach has significant implications for the comprehensive assessment of key amino acids that determine interactions necessary to support the efficient phosphoryl transfer from a protein kinase to a substrate.

A novel and rapid approach is described where ERK2 activity is monitored based on the phosphorylation of a fluorescently labeled proline-directed peptide. The unphosphorylated peptide is sensitive to protease cleavage, which releases green fluorescent protein (GFP) tag.

Phosphorylated GFP labeled peptide is stable and can be monitored by FACS analysis to determine the degree of ERK2 activity. The level of GFP signal is then correlated to different D-domain sequence mutations generated from a 6 amino acid combinatorial library.

The extensive data sets support an overall conclusion that two key hydrophobic residues in the D-domain act as an anchor and allow interactions with ERK2 regardless of other residues in D-domain. The studies are important and provide a novel approach to examine the molecular determinant of kinase interactions with substrates. However, the following items should be addressed to help complete the story.

1. Fig. 1A: Can the model discriminate between the MKK1 D-domain used in the studies and the corresponding D-domains on MKK3/4/6/7? Part of the basis for excluding the basic residues in the D-domain analysis was that they were not determinants for MAPK targeting. However, this was not examined in the current model. The use of MKK2 in Fig. 1B seemed a little confusing since the rest of the studies primarily focus on MKK1.
2. The model describes peak phosphorylation of ERK2 by MKK1 after 3 hours (Fig. S3), which seems to be quite a bit slower than typical protein kinase phosphorylation kinetics observed in other in vitro assays or cell models. Can the authors comment on this discrepancy?
3. Mutating isoleucine and leucine in MKK1 D-domain slows ERK2 phosphorylation but does not block it. This suggests other residues might be involved. Would mutating lysine/arginine residues in the D-domain block MKK1 interactions with ERK2? This relates back to point #1 and the rationale to exclude the basic residues in the D-domain.
4. The studies focus on the key residues in the D-domain of MKK1 that are important for interactions with ERK2. Conversely, the role of amino acids in the D-domain recruitment site (DRS) of ERK2 may also play a role. For example, would mutations in ERK2 DRS (eg. D321 and E322) impact MKK1 interactions?
5. Fig. S7: Panels D and E have same label, missing p1?

Reviewer #2:

Remarks to the Author:

The paper presents a droplet-based experimental method for massively parallel screening of kinase activities in a cell-free environment. The advantage of the method is clear, since it separates the target pathway from the confounding factors of a cellular environment. The authors applied this method to the MKK-ERK signaling pathway. Specifically, they made combinatorial mutagenesis library on six residues on the docking domain of MKK1, and assay the kinase activities of variants using a phosphorylation detection mechanism. The main findings include synergistic interactions among hydrophobic motifs on the docking domain, degeneracy on other positions caused by positive epistasis, and high connectivity among functional variants as a result of the high degeneracy.

Overall comment:

I think the experiment assay is very elegant and could be a valuable tool for studying sequence-function maps for kinases or other proteins. I also find the findings on the epistatic interactions interesting.

I do however, have several complaints about the statistical analyses. And I suggest that the authors redo at least some of the analyses with a more principled method to corroborate/improve

their findings, to make it suitable for publication on Nature Communications.

First, I find the the analyses very similar to Podgornia and Laub 2015. It seems to me that the authors might have used this paper as a guideline for conducting their own analyses, since Podgornia and Laub 2015 uses a very similar system and perform similar experiments (but in vivo). While this is not problematic per se, the computational methods for analyzing massive parallel mutagenesis data have advanced quite significantly in the last five years, so better, more principled analyses can be done.

Second, the inference on the per site amino acid preference and pairwise epistatic interactions are based on the enrichment ratios. Although this inference approach is common in bioinformatics, the mathematical justification for this procedure relies on multiple assumptions that are often violated in real experiments (see Atwal, G.S., Kinney, J.B.: Learning Quantitative Sequence-Function Relationships from Massively Parallel Experiments. *J Stat Phys* 162(5), 1203-1243 (2016)). In addition, there are several procedures done in the paper that might help drown out the signal with noise, or introduce biases. For example, the authors sorted the variant carrying beads into three bins, but at the end only focused on the high-activity bin, by lumping the low and medium gates together. Therefore, a lot of valuable information might have been lost this way. Next, the authors calculated the enrichment ratio for single and pairs of amino acids based the final active set of $n = 29,563$ active variants found in the high gate with counts ≥ 51 . The problem is that the per variant count data is lost in this process, so that all variants in the final pool are treated equally, regardless of their variance/noise distributions. Additionally, there could be other high activity variants that got thrown out due to low counts, which can increase the noise to signal ratio.

Based on these concerns, I suggest the authors redo the enrichment and epistatic analysis using a new software package called MAVE-NN (<https://www.biorxiv.org/content/10.1101/2020.07.14.201475v1>). MAVE-NN is a python package that is capable of learning the noise model and separates it from the biology, thus avoiding the problems with enrichment ratios. And its noise-agnostic regression functionality is specifically designed for FACs type data.

Specifically, MAVE-NN models an unobserved phenotype (in this case relative phosphorylation activity) under additive or additive + pairwise effects. It then maps the phenotype of a variant to a distribution among the gates (bins). The parameters of the model (additive or additive + pairwise regression coefficients on the phenotypic value) are then found by maximizing the likelihood of the count data.

The input to the method is the sequences and counts in each bin per sequence, so no experimental information is lost and the authors do not have to set arbitrary thresholds.

So I think most of the problems with the analyses can be solved by this package. And I have used this package before and found it very user-friendly and well documented. Therefore, I suggest that the authors try to reanalyze their data using MAVE-NN to supplement/replace the single and pairwise AA enrichment ratios with the regression coefficients from MAVE-NN.

For Figure 3C-D, use the additive noise agnostic regression in MAVE-NN.

For Figure 4, use the pairwise noise agnostic regression.

For Figure 5, the authors can generate a series of subsets of the data, where subset of sequences with vs without the motif at certain positions are fed to MAVE-NN to generate the conditional PWMs.

Detailed comments:

line 20: signalling to signaling

Line 166-171: what false positive rate? Is it 8%? The authors should point this out here.

Line 151-153: what is the criteria for selecting subsets of AA for each position? This should be

elaborated a little (perhaps in the supplement)

Fig 2D and 2G: Why do the post-digest beads have a bimodal distribution? Can the authors give a biological explanation for this in the main text?

Figure 3B. I suggest the authors add sequence logos, as the bar charts are hard to read.

Fig3C: It seems that rows 8a and I9 have been swapped.

The calculation of enrichment ratios in Fig 3 and 4 (ignore if figures are replaced with results of MAVE-NN):

Why the enrichment ratio is calculated relative to the expected frequencies in a perfectly balanced library, instead of the observed frequencies in the actual library? (What if the expected frequencies differ systematically from the frequencies in the library, due to the library preparation process?)

The pie charts in Fig 3B show a relatively balance distribution across sites in the low gate, but I do not think this justifies the use of the idealized frequencies

Fig 6: The similarity network is helpful for revealing clusters of different functional variants and their relationships. But a problem is that the variants shown here are an arbitrary subset of functional variants that might not contain ALL the functional variants due to incomplete sampling, sequence count thresholding, and experimental noise. The figure might also contain false positive variants. Therefore, I suggest the authors generate a model for the sequence-function map that contain ALL the possible variants ($n=12*13*2*12*12*12 = 539136$), and visualize the relationship among the functional variants according to this model.

For example, the authors can fit a pairwise MAVE-NN model and generate the full sequence-function map by predicting the functions for all possible sequences using the trained model, then identify the high activity subset by setting a threshold (a sensible choice is the fitted value of the WT). Alternatively, the authors could train a binary classifier using the experimental data and apply it to all possible sequences (see Podgornia and Laub 2015).

Supplement contains many typos and grammatical errors. I suggest the authors carefully read through it and fix all the problems in the revision.

Reviewer #3:

Remarks to the Author:

The authors report on (i) the development of a high throughput in vitro assay to measure mutational effects on phosphorylation activity of kinases (in a controlled context, with no inference from cellular components), and (ii) analysis of epistasis in the generated data, revealing sequence determinants of phosphorylation activity in MKK1. This study combines biochemistry, molecular biology, droplet microfluidics, high throughput sequencing, and sequence data analysis to tackle a hard problem of deep relevance both for the fundamental understanding of enzymes and biomedical applications related specifically to kinases. It thus represents an impressive amount of work of high novelty, quality, and impact. In short I find the experimental workflow and obtained results extremely interesting, and therefore in my opinion this study is suitable for publication pending major points detailed below about sequencing depth and quantitative analysis. In addition, I would suggest to emphasize the identified sequence determinants of phosphorylation activity rather than epistasis per se.

Major points regarding sequencing depth and quantitative analysis:

- The authors report data in FigS12 on wt and 56 single mutants, out of which 42 negative mutants, 1 undetermined mutant, 13 positive mutants. However the total number of possible single mutants is $12*4 + 13 + 2 = 63$ single mutants. Therefore 7 single mutants are not scored apparently, probably because of insufficient sequencing depth (too low counts), but these missing mutants might be enzymatically active (see next point). Consequently, epistasis involving these 7 mutations is more challenging to quantify a priori. Moreover, if the set of positive mutants is not robustly identified, then the argument that the exhaustive combinations of 13 mutations cannot

account for 30k observed positive variants (highlighting positive epistasis) becomes weaker. In the next point I suggest how to solve this issue.

- The definition of positive clones (Sup.Mat.) includes a criterion on the absolute number of counts in the high gate and additional criteria on the relative fractions of counts in all gates. This choice is made to avoid false positives, but ends up ranking mutants mainly according to high gate counts as shown in FigS12, which is maybe not the best choice as there are many false positives in the medium gate for instance. Aren't the relative fractions of counts in the 3 gates the most reliable indicators? For instance, PA-ILP and PW-ILP have mostly or only high gate counts but very few in absolute numbers so they are counted as negative mutants, not clear to me why (since the high gate contains very few false positives as shown in FigS4). P- -IWP is also mainly in the high gate, more so than wt, I would argue that it should be counted as positive regardless of its absolute total number of counts. Fluctuations in the absolute total number of counts between different variants may derive from lack of uniformity in the initial library (unlikely given the construction method) and/or error/biases occurring at the many steps of the workflow. My suggestion: to compute enrichments between each gate and the sequencing readout applied to the initial library, with sufficient depth to get enough counts for every library sequence. The low gate sequencing readout seems not to be a perfect proxy for the initial library sequencing readout (see below). And the ideal frequencies used to normalize observed frequencies are not a priori as reliable as actual frequencies measured in the initial library.

- The accuracy of measurements is key to quantify significant deviations from additivity of mutational effects. This is important as the experimental workflow consists in many steps (including sequencing readout), each of which adds up to the total measurement noise. What is the variance of independent data points for a single variant? (wt to start with). If I correctly understood the workflow, comparing independent measurements of the same variant in parallel batches that partially overlap should address this issue without any additional experiment. The significance of results displayed in FigS13 for instance depends critically on measurement accuracy. I am rather confident that the authors' main conclusions are correct given the wide margin they took to define active variants, and the robust Phi-X-Phi motif they report, but this important point should be addressed by further data analysis to be added to the Sup.Mat. to support the conclusions about observed epistasis.

- Fig3C data is not consistent with the structure of the library shown in Fig2B. Fig3D displays mutations that are enriched in the low gate, compared to a uniformly distributed initial library, but then the low gate is not a good proxy for the full initial library if so many mutations are significantly enriched. Please clarify this point. Does 7aI (red in Fig3C and blue in Fig3D) reflect epistasis?

- Epistasis in FigS13 is computed precisely as deviation from expectations based on independent contributions of two single amino acids to the enrichment of their pairwise combination. Aren't the enrichments of the 13 single mutants among positive variants much more accurately measured than those of the other single mutation enrichments? How does that affect computing epistasis? Wouldn't a more uniform accuracy of single mutation effects, from low activity to high activity, ensure more straightforward epistasis analysis?

Overall addressing these major points concerning the quantification of epistasis involves reanalysis of the sequencing data, and potentially deeper sequencing of current samples, but requires no additional screening experiment. However, in my opinion the study's most impactful result is not the extent of epistasis observed in the data, but rather the Phi-X-Phi motif reported by the authors, whose position is not fixed among variable positions of the library. Indeed, this finding includes the idea that epistasis plays a strong role as this sliding motif cannot be identified through a mere 'logo analysis' (Fig.3C, once corrected), and above all it captures the relation between sequence and function which is the main goal of research in the field. Several mutational scans on various proteins have reported the observation of pervasive epistasis, but very few have concluded on how function is encoded in the sequence as in the present work.

Minor points:

- the subGFP construct is described in the Sup.Mat. p.S43. I can see 3 phosphorylation sites separated by flexible linkers. Can the authors comment on this substrate structure? How was the distance between the AVI biotinylation site to the phosphorylation sites optimized in terms of chymotrypsin accessibility? Why 3 phosphorylation sites ? (reaction kinetics are potentially not 1st order)

- If I understand correctly, positive predicted value=1-false positive rate, I would just give false positive rate (in high gate) and false negative rate (in low gate, or low+med gates) in FigS4. Do FRET-tested sorted variant statistics (Fig2I) reflect what you expect from positive control and negative control sorting statistics (FigS4)? (A simple statistical test would add a quantitative argument on top of the qualitative picture provided in the text).

- The authors show a time course of the phosphorylation emulsion assay to find an optimal incubation time to discriminate between positive control and negative control. Ideally, the library whose phenotypic distribution is a priori unknown should be screened with a varying incubation time as well to maximize dynamic range and accuracy. In addition, FigS8 shows that chymotrypsin digestion readout also depends on time, so this is another parameter that could be used to maximize dynamic range and accuracy. I think this point would be worth discussing at least. Additional experiments to address this point would be asking for a lot, maybe a simulation would be possible without requiring too much time and efforts.

- FigS9 illustrates the minimization of PCR cycles to avoid biases using qPCR, which is a great idea. But then the authors should show how much of a bias there is in the first pace and how minimizing PCR cycles allowed to minimize such bias. This could be done by performing different numbers of PCR cycles upstream of sequencing and comparing results between runs. The sentence 'Plotting... correlated' in the FigS9 legend is unclear.

- FigS11 illustrates the choice of cutoff for the sequence counts to discriminate sequencing noise from signal. The choice of the cutoff is however not clearly motivated, could you plot the number of distinct sequences as a function of the cutoff and show that it becomes nearly constant above the chosen threshold? Also, the optimal cutoff is likely determined by the number of PCR cycles shown in FigS9, a simulation could help here. FigS11B is not clearly described in the legend. Figs11A, C and D: 20 should be ≥ 20 on the x-axis I guess.

- Main text line 257: 'stripes of co-enrichment between a large hydrophobic residue (Ile/Leu) at one position and any other amino acid at the other ', there is no co-enrichment proper since only one amino-acid is selected regardless of the other.

- From FigS16B, the Phi-X-Phi motif is rather Phi-X-Phi flanked by A->K amino-acids.

- caMKK1 is called both wt (Fig3D) and a constitutively active mutant (main text), potentially confusing.

- Fig3B is not so informative and could be sent to the Sup.Mat.

- Typo line 690 of the SI.

- Main text line 360: Leu-D = consensus but low affinity, a really nice illustration of epistasis. But I cannot identify Leu-D as a consensus from Fig3C, rather from top left panel of FigS15B, Fig16B, Fig17B. Is affinity synonymous with phosphorylation activity for a given peptide? (In general substrate affinity does not necessarily correlate with enzymatic activity)

- Network analysis: not sure what to take from it, could be sent to the Sup.Mat. as no clear message seems to emerge.

Lines 334-335: the authors report no hub in the network but do not show any analysis. Can you plot the node degree distribution? And the clustering coefficient (probability that neighbours are connected) vs node degree? These plots would clearly display the absence/presence of any hierarchical network structure. Also, a color heatmap of enrichments would display the enzymatic activity landscape on top of the network.

Responses to the reviewers' comments - NCOMMS-21-13316-T

We thank all reviewers for their detailed reading of the manuscript and their thoughtful and relevant comments that helped us to test and redefine our ideas and results. All attachments can be found at the bottom of this document. Attachment 1 and 2 are additionally added to the supplementary information as Supplementary Note 2, and Fig. S14.

Reviewer #1

1. *Fig. 1A: Can the model discriminate between the MKK1 D-domain used in the studies and the corresponding D-domains on MKK3/4/6/7? Part of the basis for excluding the basic residues in the D-domain analysis was that they were not determinants for MAPK targeting. However, this was not examined in the current model. The use of MKK2 in Fig. 1B seemed a little confusing since the rest of the studies primarily focus on MKK1.*

It would indeed be intriguing to follow up this excellent suggestion in future work. In this instance we have not yet used the model system to probe questions of selectivity of D-domains in the MAPK signalling pathway. Our aim was to map the sequence determinants and underlying epistasis in the MKK1 D-domain that make for complementary interactions and subsequent activation of its cognate partner, ERK2.

To clarify our objectives, we made the following change:

- Removed line 142-143: *“with recent studies emphasising either the spacer² or hydrophobic residues³ as specificity determinants for MAPK targeting.”*

Removal of this sentence might also satisfy the reviewers comment on the exclusion of basic residues, as our approach was not aimed at dissecting the individual contribution of basic/spacer and hydrophobic residues for MAPK specificity. The basic residues were additionally not randomised because:

- Bardwell et al. (DOI: 10.1074/jbc.M115.691436) reported the following; *“These results suggest that the precise chemical identity of a given basic residue does not dramatically influence binding efficiency and thus likewise does not determine binding specificity. This is consistent with the idea that the basic residues may be able to bind to corresponding acidic patches in the docking groove in a flexible or “fuzzy” manner, accounting for the lack of resolution of this portion of the D-site in many co-crystal structures”.*

Therefore, the basic residues, while important for catalytic activity (Tanoue et al. (DOI: 10.1038/35000065) and Grewal et al. (DOI: 10.1016/j.celsig.2005.04.001)) by interacting with acidic residues on the ERK2 surface, would have been very likely to be mutated either to lysine or arginine when screening for activity. Such an outcome would have been somewhat expected and thus yielded less information on epistatic relationships than randomising the other amino acids in the D-domain. Since only a limited number of residues could be chosen for randomisation, we excluded these in our design.

- When designing a future experiment, the screening capacity of the newly developed *in vitro* screen has to be considered. Randomising the basic residues would have exponentially increased the library size beyond our screening capacity (e.g. by randomising basic residues to 8 amino acids, KLH +P + 4 controls, our library size would increase 512-fold). We thought the spacer and hydrophobic residues to be of more direct interest in terms of site saturation, as a larger variety of amino acids are hydrophobic (compared to basic) and little is known about the sequence preference of spacer residues for this interaction.

There is no structure of ERK2 with an MKK1 D-domain. To make sure that the use of the co-crystallisation of MKK2's D-domain and ERK2 is not confusing the reader:

- We removed mention of MKK2 in the figure, and kept this to the figure legend. We highlighted the different amino acids that make up a typical D-domain to give context to the importance of hydrophobic residues for activity downstream. D-domain binding follows well established principles, and the figure highlights the basic requirements of these interactions (Garai 2012, DOI: 10.1126/scisignal.2003004)

2. The model describes peak phosphorylation of ERK2 by MKK1 after 3 hours (Fig. S3), which seems to be quite a bit slower than typical protein kinase phosphorylation kinetics observed in other *in vitro* assays or cell models. Can the authors comment on this discrepancy?

The reviewer's assessment of typical protein kinase phosphorylation kinetics is correct. However, the timeframe for which peak phosphorylation was found here depends both on the time needed for *in vitro* transcription and translation of MKK1 from the linear gene and the kinetics of the MKK1-ERK2 interaction. The former is thought to take up the bulk of the 3 hour incubation period, as shown by Lindenburg *et al* (DOI 10.1002/anie/202013486, Figure 3C) where formate dehydrogenase *in vitro* expression in similar droplets by (using the same *in vitro* transcription translation system) takes hours (measured by the catalytic activity in this study, reduction of NAD to NADH).

- Included in line 131: "At the optimal timepoint ($t = 3h$, which includes the time it took to express MKK1), the sequence-dependent...."

3. Mutating isoleucine and leucine in MKK1 D-domain slows ERK2 phosphorylation but does not block it. This suggests other residues might be involved. Would mutating lysine/arginine residues in the D-domain block MKK1 interactions with ERK2? This relates back to point #1 and the rationale to exclude the basic residues in the D-domain.

The reviewer's intuition is right, and other residues are indeed involved. As mentioned in point 1 of reviewer 1, the basic residues are also important for catalytic activity of MKK1 on ERK2, and it will be detrimental for said activity to mutate these residues. Truncated constitutively active MKKs without a D-domain were shown to be unable to phosphorylate ERK/p38, (Grewal *et al*, DOI: 10.1016/j.cellsig.2005.04.001) so it's very

likely that mutating all residues in the D-domain to non-preferred residues would completely block phosphorylation.

Mutation of basic amino acids could have additionally confirmed this again, but given the previously found results of Bardwell et al. (DOI: 10.1074/jbc.M115.691436) outlined in point 1, it would have been likely that all active mutants would contain mutations to a positive charge at those randomised sites, without much sequence preference for either lysine or arginine. As we were limited by our screening capacity to randomise 5-6 residues, we chose to more thoroughly randomise the spacer and hydrophobic residues, as they were considered more likely to yield interesting promiscuity in terms of sequence preference.

4. The studies focus on the key residues in the D-domain of MKK1 that are important for interactions with ERK2. Conversely, the role of amino acids in the D-domain recruitment site (DRS) of ERK2 may also play a role. For example, would mutations in ERK2 DRS (eg. D321 and E322) impact MKK1 interactions?

Previously, mutating the DRS has been shown to negatively impact ERK2 activation as well. A study by Tanoue (DOI: 10.1038/35000065) showed that mutating D321 and E322 greatly impaired the binding of ERK2 with its substrates and activators. Although a library vs library approach would be of considerable interest in the future (mutating both the D-domain and the DRS simultaneously), the current set-up was chosen because of constraints on screening capacity and difficulties constructing a randomised MKK1 and ERK2 libraries on the same paramagnetic bead.

5. Fig. S7: Panels D and E have same label, missing p1?

The panels D and E both contained unique data, and only the header was faulty:

- Changed p2 to p1 for panel D

Reviewer #2

• First, I find the the analyses very similar to Podgornia and Laub 2015. It seems to me that the authors might have used this paper as a guideline for conducting their own analyses, since Podgornia and Laub 2015 uses a very similar system and perform similar experiments (but in vivo). While this is not problematic per se, the computational methods for analyzing massive parallel mutagenesis data have advanced quite significantly in the last five years, so better, more principled analyses can be done.

The reviewer is correct in recognising that we were initially inspired by the analysis in Podgornia and Laub, though our approach to exploring epistasis in the human kinase pair differs: Podgornia and Laub highlighted specific examples of pairwise positive epistasis, while we observe formation of a hydrophobic motif through multi-dimensional epistasis.

• *Second, the inference on the per site amino acid preference and pairwise epistatic interactions are based on the enrichment ratios. Although this inference approach is common in bioinformatics, the mathematical justification for this procedure relies on multiple assumptions that are often violated in real experiments (see Atwal, G.S., Kinney, J.B.: Learning Quantitative Sequence–Function Relationships from Massively Parallel Experiments. J Stat Phys 162(5), 1203–1243 (2016)).*

We view the enrichment ratios as good indicators of epistasis, especially as this analysis was confirmed by Podgornia and Laub to indicate epistatically intertwined pairs. Although noise does affect enrichment calculations, we analyse variants of which at least (based on the PPV of the sorting gate) 91% are active. These variants include parameters of experimental noise (resulting in 9% false positives), which, by increasing the stringency of what is considered active in this group, is likely to be reduced even further.

• *In addition, there are several procedures done in the paper that might help drown out the signal with noise, or introduce biases. For example, the authors sorted the variant carrying beads into three bins, but at the end only focused on the high-activity bin, by lumping the low and medium gates together. Therefore, a lot of valuable information might have been lost this way. Next, the authors calculated the enrichment ratio for single and pairs of amino acids based the final active set of $n = 29,563$ active variants found in the high gate with counts ≥ 51 . The problem is that the per variant count data is lost in this process, so that all variants in the final pool are treated equally, regardless of their variance/noise distributions. Additionally, there could be other high activity variants that got thrown out due to low counts, which can increase the noise to signal ratio.*

We agree with reviewer 2 that the lack of information that can be derived from the exact sequencing counts results in the loss of valuable information. We chose this approach because we did not observe a correlation between activity in the secondary screen (FRET assay) and either proportion or number of sequencing reads in the medium and low gate. Consequently, we focused the analysis on the abundantly sequenced variants as a group. The ≥ 51 count arose from using WT ^{ca}MKK1 as a benchmark. We agree that this choice is conservative, intended to reduce the false positive rate in the collection of active variants.

However, we know that the high-gate sequences are active (with high accuracy), and the sequence preferences in the high gate are clear cut. We agree that the choice of the active variant threshold is ultimately arbitrary, as the exact calibration would require numerous additional FACS controls. We repeated the analysis using three different choices for the active variants (10+ reads in high gate, 51+ reads in high gate, WT-like or better distribution of reads across three gates) and found similar results (**see Attachment 1 – defining active variants**). Indeed, when the threshold of the minimal number of sequencing count for each unique variant is lowered from ≥ 51 (29,563 unique variants) to counts ≥ 10 (36,401 unique variants), adding 19% of the variants that were taken out, (as they were deemed more likely to contain false positives), the results are very similar; the same goes if instead activity is defined by the *distribution* of reads across the three FACS/NGS gates.

Currently, we kept the most conservative choice in the main text. If the reviewer requires, we could adopt this lowering of the threshold to what is considered active in the high activity gate, or switch to the definition based on % counts in each gate, although it will not affect the general conclusions drawn in the paper on basis of the high active gate variants as a group.

• Based on these concerns, I suggest the authors redo the enrichment and epistatic analysis using a new software package called MAVE-NN (<https://www.biorxiv.org/content/10.1101/2020.07.14.201475v1>). MAVE-NN is a python package that is capable of learning the noise model and separates it from the biology, thus avoiding the problems with enrichment ratios. And its noise-agnostic regression functionality is specifically designed for FACs type data.

Specifically, MAVE-NN models an unobserved phenotype (in this case relative phosphorylation activity) under additive or additive + pairwise effects. It then maps the phenotype of a variant to a distribution among the gates (bins). The parameters of the model (additive or additive + pairwise regression coefficients on the phenotypic value) are then found by maximizing the likelihood of the count data. The input to the method is the sequences and counts in each bin per sequence, so no experimental information is lost and the authors do not have to set arbitrary thresholds. So I think most of the problems with the analyses can be solved by this package. And I have used this package before and found it very user-friendly and well documented.

Therefore, I suggest that the authors try to reanalyze their data using MAVE-NN to supplement/replace the single and pairwise AA enrichment ratios with the regression coefficients from MAVE-NN.

- For Figure 3C-D, use the additive noise agnostic regression in MAVE-NN.*
- For Figure 4, use the pairwise noise agnostic regression.*
- For Figure 5, the authors can generate a series of subsets of the data, where subset of sequences with vs without the motif at certain positions are fed to MAVE-NN to generate the conditional PWMs.*

We thank the reviewer for suggesting the MAVE-NN python package and its measurement-process agnostic (MPA) regression (previously noise-agnostic regression). Our dataset (a pandas dataframe listing all sequences and FACS bin counts) is compatible with the MAVE-NN approach, although at first the application of MAVE-NN was tricky due to very limited documentation of the MPA regression. After helpful communication with the package author, these issues were resolved and we were able to test the use of MAVE-NN.

Since the full dataset is large (300K+ variants with 10 or more total reads) and heavily biased towards low activity variants (only 11% have 25% or more NGS reads in the high gate), training any of the models on the full dataset (80:20 train-test split) gave meaningless results, unless the training set was reduced and weighed in favour of the medium and high FACS bins. Unfortunately, this training dataset – constructed in line with standard practice in data science – did not give informative or indeed interpretable results under any of the three models included in the MAVE-NN package. We have attached a brief report on our MAVE-NN analysis for the reviewers' convenience (**see Attachment 3 – MAVE NN**). (We were unsure whether to include this report in the SI

– it is more a progress report on a test case rather than anything like the ‘final word’ on MAVE-NN.)

Although full use of the MAVE-NN package would have been interesting (taking the sequencing counts, and the variants sorted in low and medium gate into account), the MAVE-NN package was ultimately incompatible with our dataset, primarily because of the library design using a reduced and uneven randomisation alphabet, and because the number of reads in the low and medium gate do not inform on fitness significantly (see discussion of false positive and false negative rates in response to Reviewer 3). We have therefore opted to maintain the pooled analysis on the active sequences in the high gate to inform on sequence parameters and underlying epistasis, as this data – even without the exact sequencing counts – informs on which unique sequences constitute active MKK1 D-domains.

Detailed comments:

- *line 20: signalling to signalling*

Changed

- *Line 166-171: what false positive rate? Is it 8%? The authors should point this out here.*

The reviewer’s assessment on the false positive rate is correct, and equals (1-positive predictive value (PPV)), or 8% in, and has been spelled out in the text more clearly

- Line 143: “...which indicates a 8% false positive rate.”
- Line 166-167. Changed “we expect the positive predictive value of the high gate to be 91±2% for encoding functional D-domains” to “we expect the false positive rate of the high gate to be ~9%.”

- *Line 151-153: what is the criteria for selecting subsets of AA for each position? This should be elaborated a little (perhaps in the supplement).*

Changed at line 151-153:

“In order to stay within bounds of the screening capacity, a subset of amino acids was screened at each position, which were picked to ensure a diverse set of amino acid side chains remained to be tested”

To: In order to stay within bounds of the screening capacity, a subset of amino acids was screened at each position, which were picked to ensure amino acids with polar, charged, or hydrophobic residues were included, with a wider range of possible hydrophobic substitutions as they complement the hydrophobic pockets in the ERK2 DRS (Fig. 1B).

- *Fig 2D and 2G: Why do the post-digest beads have a bimodal distribution? Can the authors give a biological explanation for this in the main text?*

Added to line 131: *The bimodal distribution is likely to stem from the exponential activation of subGFP via activated ERK2, as the substrate (ERK2) is enzymatic, and will continue to phosphorylate subGFP upon activation by MKK1.*

- *Figure 3B. I suggest the authors add sequence logos, as the bar charts are hard to read.*

As suggested by reviewer 3, this figure has been placed in the SI.

- *Fig3C: It seems that rows 8a and 19 have been swapped.*

Reviewer is right in this assessment, the rows were indeed swapped, and the figure 3C has been remade. Thanks for spotting this mistake!

- *The calculation of enrichment ratios in Fig 3 and 4 (ignore if figures are replaced with results of MAVE-NN):*

Why the enrichment ratio is calculated relative to the expected frequencies in a perfectly balanced library, instead of the observed frequencies in the actual library? (What if the expected frequencies differ systematically from the frequencies in the library, due to the library preparation process?)

The low gate contained ~95% of the unique sequences recovered (including the inevitable false negatives, and shown as the overlap in the Venn diagram (Fig. 3A)). Analysis of these sequences revealed the low gate sequences to be close to ideal (Shannon entropy of >0.995). Additionally, we show the deviation in the low gate recovered sequences from the ideal frequencies (at 6 – 1/12, at 7a – 1/13, at 8a – 1/2, at 9, 11 and 13 – 1/12) in **Fig. S12**, and **Attachment 2 – choosing the denominator**. This figure highlights that the sequences that we recovered, are close to ideal expected frequencies, and we choose to calculate the enrichment ratio on basis of ideal frequencies expected at each position.

We agree with the reviewer that we could have calculated the enrichment ratios by dividing the frequencies observed in the high activity gate over either) those observed in the low activity gate or) all observed sequences or) the sequences with (almost) all sequencing reads in the low gate. Here, the choice of the denominator for the enrichment ratios is in some respect just as arbitrary as the definition of the active variant dataset – we did need to make a choice. We included alternative figures when dividing by the amino acid frequencies in all observed variants (**see Attachment 2 – choosing the denominator**), and the give results very close to those when dividing over ideal frequencies. The included Attachment 2 includes tables showing the ideal and experimental amino acid frequencies for both all detected and active variants, which illustrate that the latter much outweigh any differences in the denominator.

- *The pie charts in Fig 3B show a relatively balance distribution across sites in the low gate, but I do not think this justifies the use of the idealized frequencies.*

The pie charts are accompanied both by the Shannon entropy, and Fig S11 (See comment above) to establish out use of ideal frequencies. We have additionally prepared the heatmaps in which enrichments are calculated by dividing the amino acid frequencies at each position in the high gate, over the frequencies in all detected sequences (containing 91% all theoretical sequences) (**see Attachment 2 – choosing the denominator**).

- *Fig 6: The similarity network is helpful for revealing clusters of different functional variants and their relationships. But a problem is that the variants shown here are an arbitrary subset of functional variants that might not contain ALL the functional variants due to incomplete sampling, sequence count thresholding, and experimental noise.*
- *The figure might also contain false positive variants. Therefore, I suggest the authors generate a model for the sequence-function map that contain ALL the possible variants ($n=12*13*2*12*12*12 = 539136$), and visualize the relationship among the functional variants according to this model.*

For example, the authors can fit a pairwise MAVE-NN model and generate the full sequence-function map by predicting the functions for all possible sequences using the trained model, then identify the high activity subset by setting a threshold (a sensible choice is the fitted value of the WT). Alternatively, the authors could train a binary classifier using the experimental data and apply it to all possible sequences (see Podgornia and Laub 2015).

The concerns raised here are closely tied in with the issue of defining the active variant criteria and the feasibility of generating a complete sequence-function map via MAVE-NN, which have been addressed in response to previous comments. We have done our best to explore the possible methods for defining positive variants, using the WT values as suggested (using either the high gate read count or the distribution of reads across the three FACS gates). Currently, we kept the similarity network unchanged to illustrate the impact of the 8aA insertion and the clustering around the Φ -X-F motif in the network.

We have also re-examined the network when a more generous measure of activity was used (10+ reads in the high gate). This did not change the conclusions, suggesting that the data interpretation is robust and the conclusions hold regardless.

- *Supplement contains many typos and grammatical errors. I suggest the authors carefully read through it and fix all the problems in the revisions.*

We thank the reviewer for bringing this to our attention, and to the best of our knowledge, we have revised the supplement to contain as little typos and errors as possible.

Reviewer #3

Major points regarding sequencing depth and quantitative analysis:

- *The authors report data in FigS12 on wt and 56 single mutants, out of which 42 negative mutants, 1 undetermined mutant, 13 positive mutants. However the total number of possible single mutants is $12*4 + 13 + 2 = 63$ single mutants. Therefore 7*

single mutants are not scored apparently, probably because of insufficient sequencing depth (too low counts), but these missing mutants might be enzymatically active (see next point). Consequently, epistasis involving these 7 mutations is more challenging to quantify a priori. Moreover, if the set of positive mutants is not robustly identified, then the argument that the exhaustive combinations of 13 mutations cannot account for 30k observed positive variants (highlighting positive epistasis) becomes weaker. In the next point I suggest how to solve this issue.

There is a misunderstanding here regarding the expected number of mutants. Where a position is randomised to 12/13/2 possible amino acids, these divide into 11/12/1 mutant residues and 1/1/1 MKK1 WT residue per randomised positions. Therefore, we expected $11 \times 4 + 12 + 1 = 56$ single mutants, of which only one is missing due to insufficient sequencing depth. The missing mutant is also unlikely to be active, as the sequencing depth in the high gate is relatively higher (due to the low number of beads in the high gate) than in the low gate.

- The definition of positive clones (Sup.Mat.) includes a criterion on the absolute number of counts in the high gate and additional criteria on the relative fractions of counts in all gates. This choice is made to avoid false positives, but ends up ranking mutants mainly according to high gate counts as shown in FigS12, which is maybe not the best choice as there are many false positives in the medium gate for instance. Aren't the relative fractions of counts in the 3 gates the most reliable indicators? For instance, PA-ILP and PW-ILP have mostly or only high gate counts but very few in absolute numbers so they are counted as negative mutants, not clear to me why (since the high gate contains very few false positives as shown in FigS4). P- -IWP is also mainly in the high gate, more so than wt, I would argue that it should be counted as positive regardless of its absolute total number of counts. Fluctuations in the absolute total number of counts between different variants may derive from lack of uniformity in the initial library (unlikely given the construction method) and/or error/biases occurring at the many steps of the workflow.*

We thank the reviewer for his suggestions, and agree that the sequences P—IWP, PA-ILP and PW-ILP might be considered as active protein sequences. However, we know that ~9% of the variants observed in the high activity gate encode false positives. As such, while it is tempting to consider all variants which are enriched in the high gate as active, we choose a more conservative approach (in which each variant needs to be observed ≥ 51 times in the high gate). We have established that ~91% of the variants in the high activity gate are of ^{ca}MKK1-like activity. As mentioned, increasing the stringency was thought to additionally get rid of false positives, yielding a dataset of 29563 variants which were deemed active. Although it would have been of considerable interest to obtain fitness values for each individual variants, in all gates (as suggested by reviewer 2), our dataset would not permit such investigation, as discussed below under **Rev Table 1**. The 29,563 variants are of immediate interest, however, as they allowed us to determine the sequence landscape in which active D-domains exist, and proved functional in determining the underlying epistatic interactions which shape formation of the hydrophobic motif required for ERK2 activation. Nevertheless, we also considered alternative definitions of active variants, including based on the read distribution across gates and not just absolute counts (**Attachment 1 – choosing active variants**), which ultimately yield identical conclusions.

My suggestion: to compute enrichments between each gate and the sequencing readout applied to the initial library, with sufficient depth to get enough counts for every library sequence. The low gate sequencing readout seems not to be a perfect proxy for the initial library sequencing readout (see below). And the ideal frequencies used to normalize observed frequencies are not a priori as reliable as actual frequencies measured in the initial library.

While the input library is unfortunately not sequenced here, as the functionalised beads were all used in FACS and other experiments, we choose to use the ideal frequencies as a substitute. We've recovered ~91% of the theoretical library diversity. Because we found these sequences to closely represent ideal frequencies we considered the input library as ideal (see **Attachment 2 – choosing the denominator**). Alternatively, we could have calculated the enrichment based on the low gate sequences or all observed sequences, which could be considered more reliable, and has been attached for the reviewer's convenience (see **Attachment 2 – choosing the denominator**). Still, the choice does not change the biological implications derived from the dataset, especially as we focus on broad patterns, rather than individual epistatic interactions.

- The accuracy of measurements is key to quantify significant deviations from additivity of mutational effects. This is important as the experimental workflow consists in many steps (including sequencing readout), each of which adds up to the total measurement noise. What is the variance of independent data points for a single variant? (wt to start with). If I correctly understood the workflow, comparing independent measurements of the same variant in parallel batches that partially overlap should address this issue without any additional experiment. The significance of results displayed in FigS13 for instance depends critically on measurement accuracy. I am rather confident that the authors' main conclusions are correct given the wide margin they took to define active variants, and the robust Phi-X-Phi motif they report, but this important point should be addressed by further data analysis to be added to the Sup.Mat. to support the conclusions about observed epistasis.*

The variance, or spread, of two example variants, ^{ca}MKK1 and ^{ca}MKK1^{I9A/L11A} in the low, medium and high activity gate from three experiments run in parallel is shown in **Fig. S10**, and is additionally shown here in **Rev_Table 1**.

Rev_Table 1

variant	gate	% of beads			Average % beads	st. dev.	seq counts	% seq counts
caMKK1	high	55	40	55	50.0	7.1	51	42%
	med	26	30	26	27.3	1.9	36	30%
	low	17	29	17	21.0	5.7	34	28%
caMKK1_I9A/L11A	high	4.5	2.8	7.2	4.8	1.8	0	0%
	med	9.6	5.7	14	9.8	3.4	6	23%
	low	85	91	78	84.7	5.3	20	77%

The variance (or the differences in percentages of beads from either variant in the 3 different gates) all range from 1.8~7.1%. The small standard deviations suggest good data quality, i.e. reproducibility of the screening experiment. Interestingly, the variance in the average capture in each gate for both variants in parallel experiments is well correlated with the % of sequencing counts in each gate ($r^2=0.93$) (Rev_Fig 1, Fig. S10), implying little PCR bias and that NGS counts approximate the binning by function well. Although there is correlation between count and activity, we have too few gates to resolve the activity spectrum, especially for variants of medium activity, as the number of false positives increased quite substantially as a result of the spread of $caMKK1^{I9A/L11A}$ across the three gates.

So even though the spread of these two variants between parallel batches is low (and implies reproducibility of the screening experiment in parallel emulsions) the spread of $caMKK1^{I9A/L11A}$ across the three different gates (as a result of polydispersity of droplets within a single emulsion) lowers the resolution of the high gate (8% false positives, but even more so in the medium gate (26% false positives). The number of false positives is calculated as per example: for emulsion 1, the high gate contains 55% of $caMKK1$ beads, and 4.5% of $caMKK1^{I9A/L11A}$ beads. $1-(55/(55+4.5))$ yields the false discovery rate for the first emulsion. For the second emulsion, this would be $1-(40/(40+2.8))$ etc.

As the high gate suffers least from the spread of $caMKK1^{I9A/L11A}$ as a result of the polydisperse format, we choose to do our analysis on the sequences sequenced/enriched from/in this gate.

Rev_Fig 1. Correlation of variance across gates with the variance in sequencing counts across gates.

• Fig3C data is not consistent with the structure of the library shown in Fig2B. Fig3D displays mutations that are enriched in the low gate, compared to a uniformly distributed initial library, but then the low gate is not a good proxy for the full initial library if so many mutations are significantly enriched. Please clarify this point. Does 7a1 (red in Fig3C and blue in Fig3D) reflect epistasis?

We've made an error here, and the rows of I9 and 8A were accidentally swapped. Figure 3C has been corrected.

Figure 3D was unclear, and has been removed. This figure displayed all single point mutations that were enriched in the *high gate* in a dummy heatmap (from the raw sequencing counts in **Fig. S15**). This figure is therefore not displaying mutations that are enriched in the low gate, which is evenly distributed at each position for all amino acids.

• *Epistasis in FigS13 is computed precisely as deviation from expectations based on independent contributions of two single amino acids to the enrichment of their pairwise combination. Aren't the enrichments of the 13 single mutants among positive variants much more accurately measured than those of the other single mutation enrichments? How does that affect computing epistasis? Wouldn't a more uniform accuracy of single mutation effects, from low activity to high activity, ensure more straightforward epistasis analysis?*

The core of the issue here is the ability to accurately infer the enzymatic activity of any variant from the NGS read values, whether belonging to an improved variant, WT-like, non-functional or anything in-between. However, as we discussed in the response to reviewer 2, the resolution of our dataset is not sufficient to place such high confidence in individual variant measurement. Especially challenging is the interpretation of medium activity variants, which may fall below WT but still show some biological activity. Instead our experiment was calibrated with stringent controls especially for highly active variants (the boundary between the medium and high gate was set at the *median* activity of the WT variant), such that we are confident about the properties of variants sorted into the high gate. This is supported by flow cytometry controls (**Figure 2C-E, Fig. S4**).

Instead, the epistasis is not computed on basis of the expected frequency of the two single point mutations, but rather over the frequency of each separate amino acid in the total dataset of 29,563 active variants. For example, the magnitude epistasis between residue Ile at position 7a and residue Asp at position 9 would not be calculated by the pairwise deviation from both the 7aI and 9D mutants, but rather, compared to the percentage of active variants containing Ile at position 7a (14.2% of all 29,563 active variants) multiplied with the percentage of active variants containing Asp at position 9 (3.2% of all active variants). If the frequency of the pairwise interaction, or the percentage of sequences containing 7aI and 9D exceeded the expected frequency of this pair (0.142×0.320), positive epistasis is observed and vice versa.

• *Overall addressing these major points concerning the quantification of epistasis involves reanalysis of the sequencing data, and potentially deeper sequencing of current samples, but requires no additional screening experiment. However, in my opinion the study's most impactful result is not the extent of epistasis observed in the data, but rather the Phi-X-Phi motif reported by the authors, whose position is not fixed among variable positions of the library. Indeed, this finding includes the idea that epistasis plays a strong role as this sliding motif cannot be identified through a mere 'logo analysis' (Fig.3C, once corrected), and above all it captures the relation between sequence and function which is the main goal of research in the field. Several mutational scans on various proteins have reported the observation of pervasive epistasis, but very few have concluded on how function is encoded in the sequence as in the present work.*

We thank the reviewer for sharing his enthusiasm in a result that we were also excited to observe. We agree that the formation of a Phi-X-Phi motif through the interplay of positive/negative epistasis in active D-domain sequences could be highlighted more. We've removed Fig. S16B from the SI (the epistasis map of subsets containing Φ -X- Φ motifs), and added it to Main Figure 5 as Figure 5B. Figure 5A (Identical to the now removed Fig. S16A) shows as before the epistasis map of subsets containing Φ -X- Φ motifs)

- Incorporated at line 297: *“Normalising the subsets of Φ -X- Φ containing datasets for the initial distribution of amino acid frequencies in heatmap 3B gave us a glimpse into multi-dimensional epistasis i.e. the positive or negative magnitude epistasis with a third residue outside of the Φ -X- Φ motif (Figure 5B). The emergence of the distinct Φ -X- Φ motif showcases how multidimensional epistasis shapes the sequence preference for a distinct Φ -X- Φ motif in active MKK1 D-domain sequences. Whenever a Φ -X- Φ is present (may it be at positions 6-X-7a, 7a-X-9, 9-X-11 or 11-X-13) negative epistasis is observed with a third hydrophobic at any other position, while positive epistasis is observed with any third residue which is non-beneficial in isolation. Likewise, when residues other than Ile/Leu are present at these positions, positive epistasis is observed with Ile and Leu at all other positions in the subset, meaning that the frequency of finding a Ile/Leu goes beyond the already abundant frequencies of Ile and Leu at these positions in the entire active variant dataset. On the contrary, non-beneficial amino acids become ever more detrimental in this context through negative epistasis.”*

Minor points:

- *the subGFP construct is described in the Sup.Mat. p.S43. I can see 3 phosphorylation sites separated by flexible linkers. Can the authors comment on this substrate structure? How was the distance between the AVI biotinylation site to the phosphorylation sites optimized in terms of chymotrypsin accessibility? Why 3 phosphorylation sites? (reaction kinetics are potentially not 1st order)*

The three phosphorylation sites increased the sensitivity of the screen and enabled us to pick up relative activity effects for ERK2 as little as ~2 fold (incorporation of I9A and L11A (Xu, DOI: 10.1074/jbc.M102769200). The optimisation of linker sizes was not necessary, as the first design, containing the relatively standard GGSGGS linker sufficed. Although the reaction kinetics are likely affected by the additional phosphorylation sites, our model, which contains ERK2 as substrate and activator of subGFP was likely not to follow 1st order kinetics even with a single phosphorylation site.

Our screen captures the entire molecular process encompassing:

- The binding affinity of the D-domain of MKK1 to ERK2
- The precise targeting of the ERK2 DRS so that the active sites of MKK1 and ERK2 in proximity.
- Allosteric effects of the D-domain binding ERK2

- *If I understand correctly, positive predicted value=1-false positive rate, I would just give false positive rate (in high gate) and false negative rate (in low gate, or low+med*

gates) in FigS4. Do FRET-tested sorted variant statistics (Fig2I) reflect what you expect from positive control and negative control sorting statistics (FigS4)? (A simple statistical test would add a quantitative argument on top of the qualitative picture provided in the text).

The false positive rate is indeed the complement of the positive predictive value, to make this clear, the following changes were made:

- Line 143: "...which indicates a 8% false positive rate."
- Line 166-167. Changed "we expect the positive predictive value of the high gate to be 91±2% for encoding functional D-domains" to "we expect the false positive rate of the high gate to be ~9%."

With regard to a statistical test for the secondary screen, we would have had to consider an arbitrary threshold to what constitutes active compared to a non-active kinase. For instance, if we set any kinase as being active when they surpass the $caMKK1^{I9A/L11A}$ emission ratio, every kinase in the high gate would be active, and we would have a false positive rate of 0%, lower than the 8% indicated in Fig. S4. As we did not want to set an arbitrary threshold, we choose to analyse the validation in a qualitative manner.

• The authors show a time course of the phosphorylation emulsion assay to find an optimal incubation time to discriminate between positive control and negative control. Ideally, the library whose phenotypic distribution is a priori unknown should be screened with a varying incubation time as well to maximize dynamic range and accuracy. In addition, FigS8 shows that chymotrypsin digestion readout also depends on time, so this is another parameter that could be used to maximize dynamic range and accuracy. I think this point would be worth discussing at least. Additional experiments to address this point would be asking for a lot, maybe a simulation would be possible without requiring too much time and efforts.

The chymotrypsin digest was always run to completion (after 30 minutes), so that additional time would not result in additional cleavage (and has been additionally highlighted in as shown in Fig. S8). As a proxy for the bead-assay, the FRET sensor can be followed in terms of dynamic range (Rev_Fig 2, Fig. S8).

Rev_Fig 2. Time course and of the phosphorylation assay. (Left) IVTT expressed $^{ca}MKK1$ (blue) or $^{ca}MKK1^{I9A/L11A}$ was combined with purified ERK2, and the FRET sensor. After each time point, an aliquot was taken, and incubated with chymotrypsin until completion, with the resulting emission ratio plotted. (Right). End point measurement after 30 minutes incubation with FRET and ERK2, and additional 30 minutes incubation with chymotrypsin.

In the left panel, IVTT expressed $^{ca}MKK1$ or $^{ca}MKK1^{I9A/L11A}$ was mixed with the FRET sensor. At each time point, a sample was taken, and chymotrypsin was added to the mixture. The point plotted is the emission ratio after 30 minutes of chymotrypsin digestion (when it has gone to completion).

This experiment was done with different volume percentage of IVTT, and based on this experiment we chose to use a 5% (v/v) of IVTT for our subsequent FRET based assays, incubating for 30 minutes (dotted line) before chymotrypsin digest. (for Fig 2I). The comparison of addition of 5 (v/v) % $^{ca}MKK1$ or $^{ca}MKK1^{I9A/L11A}$, incubating for 30 minutes, before adding chymotrypsin and incubating for an additional 30 minutes can be seen in the right panel (and additionally in FigS8, where the end-point after 30 minutes of CT digestion results in the bar chart shown here).

Following the same rationale, when $^{ca}MKK1$ has reached saturation (having most of their beads in the subGFP positive gate after chymotrypsin digest in the droplet based assay) the dynamic range is optimal: $^{ca}MKK1^{I9A/L11A}$ will at this timepoint have phosphorylated only a fraction of the beads (false positive rate, ~8%) which are likely to arise from beads encapsulated in smaller droplets so that the effective concentration of $^{ca}MKK1^{I9A/L11A}$ is very high.

• *FigS9 illustrates the minimization of PCR cycles to avoid biases using qPCR, which is a great idea. But then the authors should show how much of a bias there is in the first pace and how minimizing PCR cycles allowed to minimize such bias. This could be done by performing different numbers of PCR cycles upstream of sequencing and comparing results between runs. The sentence 'Plotting... correlated' in the FigS9 legend is unclear.*

Unfortunately we do not have the bead samples anymore that would allow us to sequence the output after different number of cycles of amplification on the bead. Starting from the assumption that fewer cycles have lower PCR bias, we established a protocol with the minimal number of cycles necessary to reach the low-exponential threshold set it Fig S9A. We have added **Fig. S10**, which highlights the correlation of the FACS sort to the actual sequencing reads for 2 independent variants in three separate emulsions. As our analysis was done on the pool of unique variants found in the high activity gate (where each unique variant is counted as *one* of 29563 options) the PCR bias is additionally less compromising. PCR bias would be more detrimental if we tried to infer fitness for each individual variant on the basis of sequencing counts, which, due to the low resolution of especially the medium and low gate (resulting the polydisperse droplets, which increase the spread of $^{ca}MKK1$ and $^{ca}MKK1^{I9A/L11A}$ across the gates, see above), was unfortunately not possible.

Sentence in Figure legend Fig S9 changed to: “*The number of cycles required to reach the low-exponential amplification threshold (Black bar – A) was exponentially correlated to the number of beads used as template molecules in the PCR reaction.*”

• FigS11 illustrates the choice of cutoff for the sequence counts to discriminate sequencing noise from signal. The choice of the cutoff is however not clearly motivated, could you plot the number of distinct sequences as a function of the cutoff and show that it becomes nearly constant above the chosen threshold? Also, the optimal cutoff is likely determined by the number of PCR cycles shown in FigS9, a simulation could help here.

We refer the reviewer to the discussion in **Attachment 1 – choosing active variants** for an overview of our thoughts on this issue and to the response above on defining the active variants.

As requested, here we include a plot of the number of sequences as a function of the cutoff values (**Rev Fig 3**, and additionally shown in **Supplementary text 2**).

Rev Fig 3: The number of active variants as a function of the cutoff number of reads in the high gate. Highlighted are the two values (10 and 51) which are further evaluated in attachment 1.

This representation shows that there is a marked transition at ~10 high gate reads, so the cut-off should not be set lower than that value. Later the curve is continuous, so that there is no obvious choice for a 'natural' cut-off. We based our choice of the threshold value 51 on the WT read counts, but also note that alternative choices here are valid – and they lead to the same conclusions.

FigS11B is not clearly described in the legend. Figs11A, C and D: 20 should be ≥ 20 on the x-axis I guess.

The annotation in Fig. S11 (**Fig. S13 in the revised SI**) has been updated to indicate where the columns represent a sum of the variants, and the legend clarified. The discussion of variant counting was moved and expanded on in the new **Supplementary Text 2**.

Main text line 257: 'stripes of co-enrichment between a large hydrophobic residue (Ile/Leu) at one position and any other amino acid at the other', there is no co-enrichment proper since only one amino-acid is selected regardless of the other.

This is correct, and we've changed the wording to

- Line 257: "... we observed clusters of enrichment wherever a Leu/Ile is present, indicating that previously determined non-beneficial residues such as Ala are allowed in the context of a hydrophobic anchor residue.

- From FigS16B, the Phi-X-Phi motif is rather Phi-X-Phi flanked by A->K amino-acids.

Fig. S16B (the epistasis map of subsets containing Φ -X- Φ motifs) is now included in the main figures, as **Figure 5B** (the enrichment map of subsets containing Φ -X- Φ motifs). We've included a sentence that highlights the preferred A-K amino-acids.

- Figure 5 legend: "The top row shows that when the Φ -X- Φ residues are included, there is an epistatic preference for non-hydrophobic residues (Ala, Gly, Pro, Tyr, Asp or Lys) in the rest of the D-domain."

- *caMKK1* is called both wt (Fig3D) and a constitutively active mutant (main text), potentially confusing.

As described above, Figure 3D was unclear, and has been removed. This figure displayed all single point mutations that were enriched in the *high gate* in a dummy heatmap (from the raw sequencing counts in **Fig. S15**, but only added confusion to the data already present in **Fig. S15**).

- *Fig3B* is not so informative and could be sent to the Sup.Mat.

We've moved Fig3B to the Sup. Mat (**Fig.S11**). We had included it in the main figures before because we felt the low gate unique sequences indicate the quality of the library assembly, and the subsequent deviation (enrichment) in the medium and high gate.

- Typo line 690 of the SI.

Corrected

- Main text line 360: *Leu-D = consensus but low affinity, a really nice illustration of epistasis. But I cannot identify Leu-D as a consensus from Fig3C, rather from top left panel of FigS15B, Fig16B, Fig17B. Is affinity synonymous with phosphorylation activity for a given peptide? (In general substrate affinity does not necessarily correlate with enzymatic activity)*

Figure 3C (now figure 3B in the revised version) should have indeed been the same plot as the top left panels in FigS15B, Fig16B, Fig17B (now **S18B, Figure 5B, S19B**), and has been corrected. The consensus peptide therefore does match the highest enriched amino acid at each randomised position.

We've cloned each D-domain Lib1-10 as a construct with ^{ca}MKK1, replacing the wt D-domain to contain one of 10 possibilities. Each ^{ca}MKK1 construct was expressed, combining 1.25% (v/v) of the IVTT expressed variant with the FRET sensor, and measuring the emission ratio at different timepoints by taking an aliquot and incubating with chymotrypsin for 30 minutes (**Rev Fig 4**).

Rev_Fig 4. Activity of Lib1-10 D-domain-^{ca}MKK1 fusion variants over time. Each of the individually rescreened isolated D-domains Lib1-10 were cloned as fusions with ^{ca}MKK1, replacing its wildtype D-domain.

Rev_Fig 5. Correlating the affinities of Lib1-10 to their enzymatic activity as part of ^{ca}MKK1. The affinities of the rescreened D-domains Lib1-Lib10 for the DRS of ERK2 were correlated to the enzymatic activity of ^{ca}MKK1 construct with Lib1-Lib10 as its D-domain.

Each construct of ^{ca}MKK1 with one of the Lib1-10 as its D-domain is active, and will have saturated the phosphorylation cascade after ~120 minutes. The differences in activity are most pronounced after 60 minutes incubation and were correlated to the affinities of each Lib1-10 (**Rev Fig 5**). Here, the affinities of the Lib1-Lib10 D-domains showed weak correlation with their enzymatic activity, that is, the activity of ^{ca}MKK1 constructs with Lib1-Lib10 as their D-domain. Although higher affinity is weakly correlated with increase in activity, it is hardly significant ($r^2=0.2$). However, as the reviewer suggested, this is no surprise, as several studies have found only weak correlation between binding data and enzymatic activity data from high-throughput screens of kinase domains (Rudolf et al., DOI 10.1371/journal.pone.0098800).

• *Network analysis: not sure what to take from it, could be sent to the Sup.Mat. as no clear message seems to emerge.*

We view the network analysis as a showcase of positive epistasis in a highly connected network, so we would like to keep it in the main text. Furthermore, the network is in line with the expectations of multidimensional or multi-site saturation screening, as the 'holey landscape' model predicts the genotype to permeate all

regions of sequence space, underlining the importance of neutral ridges in multi-dimensional screening.

• *Lines 334-335: the authors report no hub in the network but do not show any analysis. Can you plot the node degree distribution? And the clustering coefficient (probability that neighbours are connected) vs node degree? These plots would clearly display the absence/presence of any hierarchical network structure. Also, a color heatmap of enrichments would display the enzymatic activity landscape on top of the network.*

We added the node degree distribution as **Fig. S20** and the plot of clustering coefficient vs node degree as **Fig. S21**, with an appropriate reference in the main text. These plots show that there is little hierarchical network structure in the sequence similarity network, as we had stated before.

Attachment 1: Defining the active variants: choosing the numerator.

NB. The general conclusions from attachment 1 and 2 are written up as Supplementary Note 2, whereas the heatmaps based on alternative formulations of the active dataset (numerator) divided over either ideal frequencies or the entire recovered diversity (denominator) are additionally implemented as Fig. S14.

$$\log_2 \frac{f_a^{obs}}{f_a^{fid}}$$

In this manuscript we show the enrichment ratios ($\log_2 \frac{f_a^{obs}}{f_a^{fid}}$), where the observed frequencies for active variants (f_a^{obs}) are divided by the frequencies expected in a perfect library with the ideal distribution of amino acids (f_a^{fid}). The active dataset which shapes the numerator f_a^{obs} comprises all sequences which are recovered from the high activity gate and contain >51 sequences

Here we demonstrate that the choice of filtering method for the active dataset does not change the enrichment ratios;

SINGLE ENRICHMENT

Att 1. Fig 1: Four choices for defining active variants displayed as heatmaps, with numeric value of the log2 enrichment ratio overlaid for comparison.

$$\log_2 \frac{f_a^{obs}}{f_a^{fid}}$$

This plot shows the relative observed AA distribution / ideal distribution ($\log_2 \frac{f_a^{obs}}{f_a^{fid}}$), varying the definition of 'observed' for four different ways of defining active variants. The four choices of active variants are constructed as follows:

- 10+ sequencing reads in the high gate (Att 1, Fig 1 – first heatmap): all variants that are abundant in the high gate, regardless of their appearance in the other gates. This set is the largest and the most permissive, yet it already shows a strong enrichment for L/I in most positions.
- 51+ sequencing reads in the high gate (Att 1, Fig 1 – second heatmap): the most restrictive choice, examining only variants that are at least as abundant

as WT (51 reads) in the high gate, but again ignoring the sequencing counts from the lower gates.

- WT or better distribution of reads (Att 1, Fig 1 – third heatmap): this time, the variants may be from any region of the Venn diagram (i.e. have fewer than 10 reads in the high gate), but must have <30% reads in the low gate (cf. 28% for WT) and >40% reads in the high gate (cf. WT 42%). This choice is the largest, as it also includes some rarer variants.
- The combined criterion, this time with high-gate-abundant variants (10+ reads in high gate) (Att 1, Fig 1 – fourth heatmap) that also have a suitable distribution of sequencing reads, i.e. conditions 1 and 3 combined. This option doesn't give much added value compared to each separate cut-off, while it is potentially confusing (using both absolute counts and the distribution information).

In all cases each active variant is counted only once when calculating the enrichment, i.e. the amino acid distribution is not weighed according to variant abundance within the chosen dataset.

The choice currently presented in the main manuscript (51+ reads in the high gate) shows slightly stronger preferences in positions 9 and 11, but simultaneously slightly reduces the calculated preferences in position 7a. Although it may be debated which choice is preferred, the enrichment heatmaps ultimately lead to identical conclusions.

PAIRWISE ENRICHMENT

Looking beyond the simple sequence preference in the active dataset, we also re-consider the trends in pairwise enrichment, depending on the choice of active variants.

Att 1. Fig 2: 10+ sequencing reads in the high gate (as Att 1, Fig 1 – first heatmap): all variants that are abundant in the high gate, regardless of their appearance in the other gates. This set is the largest and the most permissive, yet it already shows a strong enrichment for L/I in most positions.

Att 1. Fig 3. 51+ sequencing reads in the high gate (as Att 1, Fig 1 – second heatmap): the most restrictive choice, examining only variants that are at least as abundant as WT (51 reads) in the high gate, but again ignoring the sequencing counts from the lower gates.

Att 1. Fig 4. WT or better distribution of reads (as Att 1, Fig 1 – third heatmap): this time, the variants may be from any region of the Venn diagram (i.e. have fewer than 10 reads in the high gate), but must have <30% reads in the low gate (cf. 28% for WT) and >40% reads in the high gate (cf. WT 42%). This choice is the largest, as it also includes some rarer variants.

Attachment 2. Defining the enrichment ratios: dividing by the ideal distribution or all observed variants?

NB. The general conclusions from attachment 1 and 2 are written up as Supplementary Note 2, whereas the heatmaps based on alternative formulations of the active dataset (numerator) divided over either ideal frequencies or the entire recovered diversity (denominator) are additionally implemented as Fig. S14.

In the main manuscript we show the enrichment ratios ($\log_2 \frac{f_a^{obs}}{f_a^{fid}}$), where the observed frequency for active variants (f^{obs}) are divided by the frequencies expected in a perfect library with the ideal distribution of amino acids (f^{fid}). We chose this denominator because we did not have sequence information for the full, unsorted library.

Here, we demonstrate that this choice gives the same results as if we had chosen to use the amino acid distribution in all observed variants (f^{all}), which is our best proxy for the composition of the entire library.

Table 1: The observed proportion in % of each amino acid (f^{all}) in the combined set of all detected variants (505,957 variants).

ALL	6	7A	8A	9	11	13
IDEAL	8.333	7.692	50	8.333	8.333	8.333
A	8.473	7.509	49.624	8.354	8.340	8.448
G	8.168	7.609		7.754	8.267	8.313
P	8.225	7.751		8.455	8.235	8.358
Y	8.362	7.772		8.551	8.175	7.962
D	8.533	7.791		8.329	8.422	8.368
K	8.213	7.668		8.308	8.345	8.395
M	8.346	7.617		8.411	8.370	8.562
V	8.512	7.534		8.337	8.418	8.469
I	7.992	7.809		8.249	8.379	8.423
L	8.454	7.728		8.458	8.452	8.086
F	8.334	7.791		8.371	8.366	8.235
W	8.389	7.600		8.422	8.230	8.382
Δ		7.821	50.376			

Examining actual distribution of amino acids by position shows that the distribution is very even and close to the expected distribution (all within 0.5% absolute deviation from the ideal frequencies), but it is difficult to interpret exactly. It is easier to get a feel for the data distribution by calculating the difference between f^{all} and f^{fid} , expressed as a relative difference $(f^{all} - f^{fid})/f^{all}$. This data re-formulation is shown in Table 2 below.

Table 2: The percentage deviation from ideal amino acid frequencies, shaded by absolute deviation. 0=blue, ± 5 or more % = red.

% deviation	6	7a	8a	9	11	13
IDEAL	8.333	7.692	50	8.333	8.333	8.333
A	1.67	-2.38	-0.75	0.25	0.09	1.38
G	-1.99	-1.09		-6.95	-0.80	-0.25
P	-1.30	0.76		1.46	-1.18	0.30
Y	0.35	1.04		2.62	-1.90	-4.46
D	2.39	1.29		-0.05	1.07	0.41
K	-1.44	-0.31		-0.30	0.14	0.74
M	0.15	-0.98		0.93	0.45	2.74
V	2.15	-2.05		0.04	1.01	1.62
I	-4.10	1.51		-1.02	0.55	1.08
L	1.44	0.46		1.50	1.42	-2.96
F	0.00	1.28		0.45	0.39	-1.18
W	0.67	-1.20		1.07	-1.24	0.59
Δ		1.68	0.75			

Most amino acid frequencies are within 2% relative deviation of the ideal frequency, with only three exceptions:

- Glycine at position 9,
- Isoleucine at position 6,
- Tyrosine at position 13,

All of which are mildly depleted (=underrepresented) from the overall library. That has the practical effect of reducing the signal/noise ratio for these amino acids if they were enriched, and increasing S/N for the depletion. However, these values should be considered in tandem with the size of changes we observe in the active variants:

Table 3: The amino acid frequencies by position in the active variants

Active	6	7a	8a	9	11	13
IDEAL	8.333	7.692	50	8.333	8.333	8.333
A	5.451	4.380	50.222	4.995	3.863	7.067
G	6.263	4.363		3.126	3.476	4.914
P	11.227	9.558		6.925	4.160	7.114
Y	6.627	3.994		4.922	4.419	5.426
D	2.962	1.822		3.393	3.184	5.120
K	7.578	6.953		4.705	3.651	5.882
M	9.341	5.345		10.215	7.829	9.564
V	6.652	7.350		6.399	8.001	10.123
I	7.631	14.890		14.586	19.311	13.185
L	15.009	21.956		22.515	23.338	10.902
F	10.760	7.442		10.184	10.498	9.575
W	10.498	5.740		8.034	8.268	11.130
Δ		6.207	49.778			

Here, the deviations are substantial, reaching 4-fold or more (except for position 8a); the deviations are further highlighted in a relative table with colour shading:

Table 4: The relative deviation from ideal amino acid frequency in active variants

	% deviation					
	6	7a	8a	9	11	13
IDEAL	8.333	7.692	50	8.333	8.333	8.333
A	-34.59	-43.06	0.44	-47.44	-53.65	-15.20
G	-24.85	-43.28		-47.64	-58.28	-41.03
P	34.72	24.26		14.70	-50.08	-14.63
Y	-20.47	-48.08		-52.08	-46.97	-34.89
D	-64.46	-76.32		-78.14	-61.79	-38.56
K	-9.06	-9.62		-16.57	-56.18	-29.42
M	12.10	-30.51		-35.86	-6.06	14.77
V	-20.17	-4.45		-11.80	-3.99	21.48
I	-8.43	93.56		78.68	131.74	58.22
L	80.11	185.43		163.47	180.06	30.82
F	29.12	-3.25		-10.70	25.98	14.90
W	25.98	-25.38		-31.12	-0.78	33.56
Δ		-19.31	-0.44			

Consequently, while we observe very minor deviations from ideal amino acid distribution in the set of all detected variants, these differences are trumped by the changes in amino acid distribution in the active set. When the data is cast into a relative enrichment format (see below), the results are visually identical for the plots where the different active datasets (see **Attachment 1**) are divided over the ideal frequencies or over the amino acid frequencies in all variants.

Att 2. Fig 2: Three choices of active variants, with enrichment ratios defined over the ideal library distribution or all observed SPiMLiB variants. The $\log_2(\text{ratio})$ is superimposed on the heatmaps to illustrate the small numeric difference. Included in the SI as Fig. S14

As such we believe our choice to present enrichment factors as active/ideal in the main manuscript as justified. Furthermore, there is also no discernible difference on examination of the 2D enrichment plots:

Att 2. Fig2: Pairwise enrichment plots for enrichment ratios calculated between the distribution in active variants (>51 reads in high gate, as in the main text) compared to the ideal distribution; Figure 4 in main text.

Attachment 3: MAVE-NN

The reviewers requested that we use the MAVE-NN package to interpret out FACS-NGS scores, instead of presenting a statistical analysis of the active kinase variants as a group. Any analysis is based on the following evidence derived from the FACS-NGS process: each variant has a genotype, phenotype, the distribution across FACS bins and the associated NGS read distribution, such that each step in the process is both a direct reflection of the previous steps and influenced by the noise in the process.

The ambition of MAVE-NN is to work backwards: 1) inferring phenotypes from the NGS distribution, then 2) interpreting the genotype-phenotype map to distinguish between global and local effects.

In order to take advantage of this process, the training set for MAVE-NN has to include active variants, inactive variants and everything in-between. Here, the trial MAVE-NN dataset is constructed with 386K variants that have at least 10 reads across the three gates. The emphasis is not on the exact set of variants to include (the borderlines are from the low set anyway), but on obtaining a valid estimation of variant phenotypes from the NGS data.

Matplotlib is building the font cache; this may take a moment.

First, the SpliMLiB dataset is imported and the header displayed to check formatting and columns names.

	mutations	ct_2	ct_1	ct_0	sequences
0	6L/7aI/8aA/9L/11F/13M	1185.0	33.0	6.0	LIALFM
1	6F/7aP/9W/11L/13M	903.0	81.0	15.0	FPΔWLM
2	6L/7aF/9L/11I/13I	899.0	2.0	8.0	LFΔLII
3	6A/7aI/8aA/9L/11L/13I	893.0	3.0	19.0	AIALLI
4	6W/7aI/9F/11L/13V	880.0	34.0	20.0	WIΔFLV

The column headers are as follows:

- mutation: each variant listing the mutation from WT caMKK1. Note that changes at positions 7a and 8a are listed only if there is an insertion present.
- ct_2: count of sequencing reads in the high gate
- ct_1: sequencing reads in medium gate
- ct_0: sequencing reads in low gate
- sequences: the amino acid residues in the six randomised positions, without intervening constant residues.

This format is compatible with MAVE-NN.

Next, generate some descriptive statistics of this dataset.

	ct_2	ct_1	ct_0
count	386031.000000	386031.000000	386031.000000
mean	12.008556	7.701433	15.616425
std	45.869569	12.625967	8.688312
min	0.000000	0.000000	0.000000
25%	0.000000	0.000000	10.000000
50%	0.000000	1.000000	14.000000
75%	0.000000	12.000000	20.000000
max	1185.000000	165.000000	95.000000

The signal from the NGS is normalised to convert into percentage of reads in each of the three NGS gates: these are now listed in columns 'high', 'med' and 'low'.

	mutations	ct_2	ct_1	ct_0	sequences	high	med	low
0	6L/7aI/8aA/9L/11F/13M	1185.0	33.0	6.0	LIALFM	96.814	2.696	0.490
1	6F/7aP/9W/11L/13M	903.0	81.0	15.0	FPΔWLM	90.390	8.108	1.502
2	6L/7aF/9L/11I/13I	899.0	2.0	8.0	LFΔLII	98.900	0.220	0.880
3	6A/7aI/8aA/9L/11L/13I	893.0	3.0	19.0	AIALLI	97.596	0.328	2.077
4	6W/7aI/9F/11L/13V	880.0	34.0	20.0	WIΔFLV	94.218	3.640	2.141

In our hands attempts at getting a sensible result from the MAVE-NN were unsuccessful with random sampling of this large set of variants, whether using NGS counts or the percentage distribution between bins. We suspect the likely reason for this failure is that the full dataset is strongly biased towards low-activity variants (as most mutants are compromised in their function and were sorted in the low gate) - consequently all models just allocate all variants to a low-activity phenotype. A realistic model should instead predict that the numerically estimated latent phenotype is very negative for low gate sequences, rising for variants abundant in the high gate.

To remedy this situation, we constructed a stratified training set, which creates unequal test-train splits in three different sections. Here, it is not truly important where the boundaries between the groups are (certainly the "high" group is fairly permissive), but that the relative group sizes are approximately balanced. Specifically:

- high (high > 25): place 80% in training set
- mostly medium (high < 25, med: place 10% in training set
- very low (high < 1, med < 10): place 5% in training set
- everything else: place 10% in training set

High: 38820

Low to medium: 143504

Very low: 182457

Checking that the final constructed dataset makes sense:

	mutations	ct_2	ct_1	ct_0	sequences	high	med	low	str
0	6L/7aI/8aA/9L/11F/13M	1185.0	33.0	6.0	LIALFM	96.814	2.696	0.490	
1	6F/7aP/9W/11L/13M	903.0	81.0	15.0	FPΔWLM	90.390	8.108	1.502	
2	6L/7aF/9L/11I/13I	899.0	2.0	8.0	LFΔLII	98.900	0.220	0.880	
3	6A/7aI/8aA/9L/11L/13I	893.0	3.0	19.0	AIALLI	97.596	0.328	2.077	
4	6W/7aI/9F/11L/13V	880.0	34.0	20.0	WIΔFLV	94.218	3.640	2.141	
...
386026	6P/7aL/9P/11G/13K	0.0	0.0	10.0	PLΔPGK	0.000	0.000	100.000	
386027	6M/7aV/8aA/9Y/11P/13W	0.0	0.0	10.0	MVAYPW	0.000	0.000	100.000	
386028	6L/7aW/9Y/11I/13G	0.0	0.0	10.0	LWΔYIG	0.000	0.000	100.000	
386029	6G/7aP/8aA/9G/11Y/13V	0.0	0.0	10.0	GPAGYV	0.000	0.000	100.000	
386030	6D/7aK/9M/11G/13K	0.0	0.0	10.0	DKΔMGK	0.000	0.000	100.000	

386031 rows × 10 columns

Next, we set up the data in a format fitting with the MAVE-NN package, following the original code and adjusting for the measurement-agnostic process (which has little documentation yet). The test and train set are constant for all variation of the G-P map.

Training set size: 56,591 observations
 Test set size : 329,440 observations

Additive model

N = 56,591 observations set as training data.
Using 19.9% for validation.
Data shuffled.
Time to set data: 0.313 sec.

LSMR Least-squares solution of $Ax = b$

The matrix A has 45332 rows and 78 cols
damp = 0.000000000000000e+00

atol = 1.00e-06 conlim = 1.00e+08

btol = 1.00e-06 maxiter = 78

itn	x(1)	norm r	norm Ar	compatible	LS	norm A	c
cond A							
0	0.00000e+00	2.521e+02	2.773e+04	1.0e+00	4.4e-01		
1	2.66507e-01	1.595e+02	2.162e+03	6.3e-01	9.5e-02	1.4e+02	1.
0e+00							
2	1.48419e-01	1.560e+02	3.894e+02	6.2e-01	1.6e-02	1.6e+02	2.
2e+00							
3	1.50647e-01	1.558e+02	1.845e+02	6.2e-01	7.2e-03	1.7e+02	2.
8e+00							
4	1.22148e-01	1.557e+02	1.137e+02	6.2e-01	4.2e-03	1.7e+02	3.
3e+00							
5	1.10826e-01	1.557e+02	9.270e+01	6.2e-01	3.3e-03	1.8e+02	4.
7e+00							
6	8.96984e-02	1.557e+02	7.973e+01	6.2e-01	2.7e-03	1.9e+02	6.
8e+00							
7	5.30634e-02	1.556e+02	5.720e+01	6.2e-01	1.8e-03	2.0e+02	6.
1e+00							
8	1.61922e-02	1.556e+02	2.470e+01	6.2e-01	7.6e-04	2.1e+02	4.
3e+00							
9	9.92092e-03	1.556e+02	7.926e+00	6.2e-01	2.3e-04	2.2e+02	4.
3e+00							
10	9.16781e-03	1.556e+02	2.084e+00	6.2e-01	5.9e-05	2.3e+02	4.
3e+00							
14	9.09072e-03	1.556e+02	4.584e-02	6.2e-01	1.0e-06	2.9e+02	4.
3e+00							
15	9.09104e-03	1.556e+02	1.322e-02	6.2e-01	2.9e-07	3.0e+02	4.
3e+00							

LSMR finished

The least-squares solution is good enough, given atol

istop = 2 normr = 1.6e+02
normA = 3.0e+02 normAr = 1.3e-02
itn = 15 condA = 4.3e+00
normx = 1.6e+00
15 9.09104e-03 1.556e+02 1.322e-02
6.2e-01 2.9e-07 3.0e+02 4.3e+00

Linear regression time: 0.0172 sec

Epoch 1/1000

91/91 [=====] - 0s 3ms/step - loss: 51997.1523 -
I_var: -0.0059 - val_loss: 51527.0703 - val_I_var: -0.0059

Epoch 2/1000

91/91 [=====] - 0s 2ms/step - loss: 51836.1992 -
I_var: -0.0014 - val_loss: 51462.3594 - val_I_var: -0.0039

Epoch 3/1000

91/91 [=====] - 0s 2ms/step - loss: 51808.0938 -
I_var: -8.2893e-04 - val_loss: 51472.1367 - val_I_var: -0.0044

Epoch 4/1000

91/91 [=====] - 0s 2ms/step - loss: 51796.1445 -
I_var: -3.5329e-04 - val_loss: 51449.8789 - val_I_var: -0.0036

Epoch 5/1000

91/91 [=====] - 0s 2ms/step - loss: 51797.6250 -
I_var: -2.7909e-04 - val_loss: 51443.1250 - val_I_var: -0.0036

Epoch 6/1000

91/91 [=====] - 0s 2ms/step - loss: 51758.4922 -

I_var: 4.1321e-04 - val_loss: 51438.9688 - val_I_var: -0.0034
Epoch 7/1000
91/91 [=====] - 0s 2ms/step - loss: 51765.4961 -
I_var: 6.3148e-04 - val_loss: 51449.0469 - val_I_var: -0.0037
Epoch 8/1000
91/91 [=====] - 0s 2ms/step - loss: 51758.7930 -
I_var: 7.1712e-04 - val_loss: 51440.5234 - val_I_var: -0.0035
Epoch 9/1000
91/91 [=====] - 0s 2ms/step - loss: 51768.8945 -
I_var: 3.8083e-04 - val_loss: 51428.3555 - val_I_var: -0.0032
Epoch 10/1000
91/91 [=====] - 0s 2ms/step - loss: 51759.3594 -
I_var: 5.8253e-04 - val_loss: 51456.7148 - val_I_var: -0.0041
Epoch 11/1000
91/91 [=====] - 0s 2ms/step - loss: 51760.8672 -
I_var: 6.0114e-04 - val_loss: 51435.9844 - val_I_var: -0.0035
Epoch 12/1000
91/91 [=====] - 0s 2ms/step - loss: 51757.2969 -
I_var: 5.8856e-04 - val_loss: 51431.7461 - val_I_var: -0.0033
Epoch 13/1000
91/91 [=====] - 0s 2ms/step - loss: 51763.5312 -
I_var: 2.7021e-04 - val_loss: 51428.9766 - val_I_var: -0.0034
Epoch 14/1000
91/91 [=====] - 0s 2ms/step - loss: 51754.9258 -
I_var: 5.5218e-04 - val_loss: 51429.8359 - val_I_var: -0.0033
Epoch 15/1000
91/91 [=====] - 0s 2ms/step - loss: 51764.2383 -
I_var: 4.2085e-04 - val_loss: 51448.3867 - val_I_var: -0.0040
Epoch 16/1000
91/91 [=====] - 0s 2ms/step - loss: 51768.0703 -
I_var: 2.7098e-04 - val_loss: 51476.6797 - val_I_var: -0.0048
Epoch 17/1000
91/91 [=====] - 0s 2ms/step - loss: 51768.9570 -
I_var: 1.5005e-04 - val_loss: 51465.7070 - val_I_var: -0.0045
Epoch 18/1000
91/91 [=====] - 0s 2ms/step - loss: 51751.6055 -
I_var: 5.5591e-04 - val_loss: 51423.5977 - val_I_var: -0.0034
Epoch 19/1000
91/91 [=====] - 0s 2ms/step - loss: 51758.8984 -
I_var: 4.2961e-04 - val_loss: 51425.3008 - val_I_var: -0.0033
Epoch 20/1000
91/91 [=====] - 0s 2ms/step - loss: 51758.5898 -
I_var: 5.3183e-04 - val_loss: 51428.7422 - val_I_var: -0.0035
Epoch 21/1000
91/91 [=====] - 0s 2ms/step - loss: 51759.3281 -
I_var: 3.9071e-04 - val_loss: 51431.7148 - val_I_var: -0.0036
Epoch 22/1000
91/91 [=====] - 0s 2ms/step - loss: 51752.0430 -
I_var: 5.4341e-04 - val_loss: 51448.1172 - val_I_var: -0.0041
Epoch 23/1000
91/91 [=====] - 0s 2ms/step - loss: 51754.6406 -
I_var: 4.3641e-04 - val_loss: 51461.8945 - val_I_var: -0.0046
Epoch 24/1000
91/91 [=====] - 0s 2ms/step - loss: 51745.0195 -
I_var: 4.8812e-04 - val_loss: 51416.8516 - val_I_var: -0.0032
Epoch 25/1000
91/91 [=====] - 0s 2ms/step - loss: 51755.4453 -
I_var: 6.2043e-04 - val_loss: 51435.7305 - val_I_var: -0.0037
Epoch 26/1000
91/91 [=====] - 0s 2ms/step - loss: 51748.4297 -
I_var: 5.8334e-04 - val_loss: 51486.9219 - val_I_var: -0.0053
Epoch 27/1000
91/91 [=====] - 0s 2ms/step - loss: 51745.0430 -
I_var: 6.5181e-04 - val_loss: 51444.1484 - val_I_var: -0.0041
Epoch 28/1000
91/91 [=====] - 0s 2ms/step - loss: 51738.8359 -
I_var: 6.6955e-04 - val_loss: 51426.6289 - val_I_var: -0.0036
Epoch 29/1000
91/91 [=====] - 0s 2ms/step - loss: 51747.6523 -

```

I_var: 4.9691e-04 - val_loss: 51425.4883 - val_I_var: -0.0036
Epoch 30/1000
91/91 [=====] - 0s 2ms/step - loss: 51737.3008 -
I_var: 8.0747e-04 - val_loss: 51429.5117 - val_I_var: -0.0037
Epoch 31/1000
91/91 [=====] - 0s 2ms/step - loss: 51741.9688 -
I_var: 7.1090e-04 - val_loss: 51452.0078 - val_I_var: -0.0044
Epoch 32/1000
91/91 [=====] - 0s 2ms/step - loss: 51731.6172 -
I_var: 8.7267e-04 - val_loss: 51421.7344 - val_I_var: -0.0035
Epoch 33/1000
91/91 [=====] - 0s 2ms/step - loss: 51731.8516 -
I_var: 9.4774e-04 - val_loss: 51428.8242 - val_I_var: -0.0038
Epoch 34/1000
91/91 [=====] - 0s 2ms/step - loss: 51734.2852 -
I_var: 7.3793e-04 - val_loss: 51433.1719 - val_I_var: -0.0038
Training time: 6.3 seconds
Model saved to these files:
    splimlib_mpa_additive.pickle
    splimlib_mpa_additive.h5

```

```

Model loaded from these files:
    splimlib_mpa_additive.pickle
    splimlib_mpa_additive.h5

```

The left-hand plot is a display of the measurement process (estimating the true phenotype from the NGS distribution). The right-hand plot is a plot of the genotype-phenotype map, showing how the model interprets the link between the amino acids at one sequence position, and the fitness.

Looking at the measurement map, there is a mild trend in the correct direction: variants abundant in bin 0 (low) have a negative phenotype, while variants abundant in bin 2 (high) have a positive phenotype. This trend does seem to suffer from much noise, compared to Figure 2 and Figure 5 in the preprint that describes MAVE-NN (<https://www.biorxiv.org/content/10.1101/2020.07.14.201475v3> (<https://www.biorxiv.org/content/10.1101/2020.07.14.201475v3>)).

Neighbour epistasis model

Allowing epistasis only between neighbouring positions.

N = 56,591 observations set as training data.
Using 20.3% for validation.
Data shuffled.
Time to set data: 0.29 sec.

LSMR Least-squares solution of $Ax = b$

The matrix A has 45095 rows and 78 cols
damp = 0.00000000000000e+00

atol = 1.00e-06 conlim = 1.00e+08

btol = 1.00e-06 maxiter = 78

itn	x(1)	norm r	norm Ar	compatible	LS	norm A	c
cond A							
0	0.00000e+00	2.511e+02	2.751e+04	1.0e+00	4.4e-01		
1	2.64950e-01	1.592e+02	2.285e+03	6.3e-01	1.0e-01	1.4e+02	1.
0e+00							
2	1.20690e-01	1.553e+02	4.323e+02	6.2e-01	1.8e-02	1.6e+02	2.
2e+00							
3	1.33290e-01	1.551e+02	2.046e+02	6.2e-01	7.9e-03	1.7e+02	2.
6e+00							
4	1.04756e-01	1.550e+02	1.133e+02	6.2e-01	4.1e-03	1.8e+02	3.
0e+00							
5	9.27031e-02	1.550e+02	8.612e+01	6.2e-01	3.0e-03	1.8e+02	4.
1e+00							
6	7.79318e-02	1.550e+02	7.413e+01	6.2e-01	2.5e-03	1.9e+02	6.
3e+00							
7	5.16626e-02	1.549e+02	5.913e+01	6.2e-01	1.9e-03	2.0e+02	6.
5e+00							
8	1.25131e-02	1.549e+02	2.773e+01	6.2e-01	8.5e-04	2.1e+02	4.
3e+00							
9	1.88797e-03	1.549e+02	7.812e+00	6.2e-01	2.3e-04	2.2e+02	3.
9e+00							
10	1.08035e-03	1.549e+02	2.220e+00	6.2e-01	6.3e-05	2.3e+02	3.
9e+00							
14	1.02003e-03	1.549e+02	4.611e-02	6.2e-01	1.0e-06	2.9e+02	3.
9e+00							
15	1.01832e-03	1.549e+02	1.419e-02	6.2e-01	3.1e-07	3.0e+02	3.
9e+00							

LSMR finished

The least-squares solution is good enough, given atol

istop = 2 normr = 1.5e+02
normA = 3.0e+02 normAr = 1.4e-02
itn = 15 condA = 3.9e+00
normx = 1.6e+00
15 1.01832e-03 1.549e+02 1.419e-02
6.2e-01 3.1e-07 3.0e+02 3.9e+00

Linear regression time: 0.0127 sec

Epoch 1/1000

91/91 [=====] - 1s 8ms/step - loss: 52100.4414 -
I_var: -0.0074 - val_loss: 52216.6836 - val_I_var: -0.0091

Epoch 2/1000

91/91 [=====] - 1s 7ms/step - loss: 51891.9844 -
I_var: -0.0027 - val_loss: 51969.7773 - val_I_var: -0.0021

Epoch 3/1000

91/91 [=====] - 1s 6ms/step - loss: 51822.5000 -
I_var: 3.5852e-05 - val_loss: 52229.6953 - val_I_var: -0.0096

Epoch 4/1000

91/91 [=====] - 1s 8ms/step - loss: 51789.0234 -
I_var: 0.0014 - val_loss: 51992.9062 - val_I_var: -0.0028

Epoch 5/1000

91/91 [=====] - 1s 7ms/step - loss: 51734.6641 -
I_var: 0.0029 - val_loss: 51928.7422 - val_I_var: -9.7752e-04

Epoch 6/1000

91/91 [=====] - 1s 6ms/step - loss: 51739.6992 -

```
I_var: 0.0025 - val_loss: 52025.8203 - val_I_var: -0.0038
Epoch 7/1000
91/91 [=====] - 1s 7ms/step - loss: 51726.2930 -
I_var: 0.0029 - val_loss: 51916.5859 - val_I_var: -6.9225e-04
Epoch 8/1000
91/91 [=====] - 1s 7ms/step - loss: 51695.9688 -
I_var: 0.0038 - val_loss: 51969.7539 - val_I_var: -0.0023
Epoch 9/1000
91/91 [=====] - 1s 6ms/step - loss: 51699.2969 -
I_var: 0.0035 - val_loss: 52034.8320 - val_I_var: -0.0042
Epoch 10/1000
91/91 [=====] - 1s 6ms/step - loss: 51680.7070 -
I_var: 0.0048 - val_loss: 51996.2852 - val_I_var: -0.0031
Epoch 11/1000
91/91 [=====] - 1s 7ms/step - loss: 51664.0312 -
I_var: 0.0043 - val_loss: 52100.8320 - val_I_var: -0.0061
Epoch 12/1000
91/91 [=====] - 1s 7ms/step - loss: 51667.7695 -
I_var: 0.0044 - val_loss: 52011.1562 - val_I_var: -0.0036
Epoch 13/1000
91/91 [=====] - 1s 7ms/step - loss: 51648.4414 -
I_var: 0.0048 - val_loss: 51903.3867 - val_I_var: -4.7488e-04
Epoch 14/1000
91/91 [=====] - 1s 7ms/step - loss: 51649.0938 -
I_var: 0.0045 - val_loss: 52002.5156 - val_I_var: -0.0034
Epoch 15/1000
91/91 [=====] - 1s 7ms/step - loss: 51641.5195 -
I_var: 0.0047 - val_loss: 52237.6836 - val_I_var: -0.0102
Epoch 16/1000
91/91 [=====] - 1s 7ms/step - loss: 51654.4805 -
I_var: 0.0046 - val_loss: 52100.0234 - val_I_var: -0.0062
Epoch 17/1000
91/91 [=====] - 1s 7ms/step - loss: 51633.9609 -
I_var: 0.0046 - val_loss: 52094.9648 - val_I_var: -0.0061
Epoch 18/1000
91/91 [=====] - 1s 8ms/step - loss: 51644.7383 -
I_var: 0.0053 - val_loss: 51970.9844 - val_I_var: -0.0026
Epoch 19/1000
91/91 [=====] - 1s 7ms/step - loss: 51639.9727 -
I_var: 0.0050 - val_loss: 51970.5547 - val_I_var: -0.0026
Epoch 20/1000
91/91 [=====] - 1s 7ms/step - loss: 51628.4141 -
I_var: 0.0054 - val_loss: 52068.0742 - val_I_var: -0.0054
Epoch 21/1000
91/91 [=====] - 1s 6ms/step - loss: 51619.6094 -
I_var: 0.0060 - val_loss: 51920.7070 - val_I_var: -0.0012
Epoch 22/1000
91/91 [=====] - 1s 7ms/step - loss: 51619.4609 -
I_var: 0.0053 - val_loss: 52106.2383 - val_I_var: -0.0065
Epoch 23/1000
91/91 [=====] - 1s 6ms/step - loss: 51616.9727 -
I_var: 0.0054 - val_loss: 52078.4844 - val_I_var: -0.0057
Training time: 17.5 seconds
Model saved to these files:
    splimlib_mpa_neighbour.pickle
    splimlib_mpa_neighbour.h5
```

Model loaded from these files:
splimlib_mpa_neighbour.pickle
splimlib_mpa_neighbour.h5

As in the additive model, the measurement process shows the right trend, although the spread in the colouring is still substantial.

Given our experience with the analysis of the active variants, this model should capture at least some epistasis in the G-P map (the two hydrophobics motif). However, the G-P map is rather impossible to interpret in a meaningful manner:

- the model suggests a preference deletions in positions where no deletion were, in fact, introduced to the library
- there is a preference of I/L in position 7a (1 on chart), but not in 9/11 (3/4)
- it places K, P and D residues as contributing to activity in position 11 (4 on chart), when an I/L residue in fact strongly predominates in the active variants

Pairwise model

Allowing epistasis between all pairs of positions (first order epistasis across the domain).

N = 56,591 observations set as training data.
Using 19.9% for validation.
Data shuffled.
Time to set data: 0.273 sec.

LSMR Least-squares solution of $Ax = b$

The matrix A has 45321 rows and 78 cols
damp = 0.00000000000000e+00

atol = 1.00e-06 conlim = 1.00e+08

btol = 1.00e-06 maxiter = 78

itn	x(1)	norm r	norm Ar	compatible	LS	norm A	c
cond A							
0	0.00000e+00	2.520e+02	2.771e+04	1.0e+00	4.4e-01		
1	2.57861e-01	1.595e+02	2.174e+03	6.3e-01	9.6e-02	1.4e+02	1.
0e+00							
2	1.41338e-01	1.559e+02	4.134e+02	6.2e-01	1.7e-02	1.6e+02	2.
2e+00							
3	1.38557e-01	1.557e+02	1.921e+02	6.2e-01	7.5e-03	1.7e+02	2.
8e+00							
4	1.09604e-01	1.557e+02	1.198e+02	6.2e-01	4.4e-03	1.7e+02	3.
3e+00							
5	9.79046e-02	1.556e+02	9.743e+01	6.2e-01	3.4e-03	1.8e+02	4.
6e+00							
6	7.75371e-02	1.556e+02	8.426e+01	6.2e-01	2.8e-03	1.9e+02	6.
7e+00							
7	4.12142e-02	1.555e+02	6.317e+01	6.2e-01	2.0e-03	2.0e+02	6.
1e+00							
8	1.64427e-03	1.555e+02	2.749e+01	6.2e-01	8.4e-04	2.1e+02	4.
0e+00							
9	-7.48837e-03	1.555e+02	7.966e+00	6.2e-01	2.3e-04	2.2e+02	4.
0e+00							
10	-8.25086e-03	1.555e+02	1.999e+00	6.2e-01	5.7e-05	2.3e+02	4.
0e+00							
14	-8.28108e-03	1.555e+02	4.334e-02	6.2e-01	1.0e-06	2.8e+02	4.
0e+00							
15	-8.27932e-03	1.555e+02	1.366e-02	6.2e-01	3.0e-07	3.0e+02	4.
0e+00							

LSMR finished

The least-squares solution is good enough, given atol

istop = 2 normr = 1.6e+02
normA = 3.0e+02 normAr = 1.4e-02
itn = 15 condA = 4.0e+00
normx = 1.7e+00
15 -8.27932e-03 1.555e+02 1.366e-02
6.2e-01 3.0e-07 3.0e+02 4.0e+00

Linear regression time: 0.0237 sec

Epoch 1/1000

91/91 [=====] - 1s 10ms/step - loss: 52061.5898
- I_var: -0.0079 - val_loss: 51567.2656 - val_I_var: -0.0068

Epoch 2/1000

91/91 [=====] - 1s 7ms/step - loss: 51743.7617 -
I_var: 0.0013 - val_loss: 51521.0547 - val_I_var: -0.0056

Epoch 3/1000

91/91 [=====] - 1s 7ms/step - loss: 51589.6953 -
I_var: 0.0058 - val_loss: 51985.6562 - val_I_var: -0.0190

Epoch 4/1000

91/91 [=====] - 1s 8ms/step - loss: 51545.3984 -
I_var: 0.0069 - val_loss: 51634.5117 - val_I_var: -0.0088

Epoch 5/1000

91/91 [=====] - 1s 7ms/step - loss: 51440.1992 -
I_var: 0.0097 - val_loss: 51582.1641 - val_I_var: -0.0073

Epoch 6/1000

91/91 [=====] - 1s 7ms/step - loss: 51386.6562 -

```

I_var: 0.0113 - val_loss: 51748.7266 - val_I_var: -0.0122
Epoch 7/1000
91/91 [=====] - 1s 7ms/step - loss: 51340.3398 -
I_var: 0.0127 - val_loss: 51570.3867 - val_I_var: -0.0070
Epoch 8/1000
91/91 [=====] - 1s 8ms/step - loss: 51309.1758 -
I_var: 0.0138 - val_loss: 51962.1562 - val_I_var: -0.0181
Epoch 9/1000
91/91 [=====] - 1s 7ms/step - loss: 51247.0039 -
I_var: 0.0152 - val_loss: 51865.6406 - val_I_var: -0.0153
Epoch 10/1000
91/91 [=====] - 1s 7ms/step - loss: 51243.7891 -
I_var: 0.0156 - val_loss: 51742.1250 - val_I_var: -0.0118
Epoch 11/1000
91/91 [=====] - 1s 8ms/step - loss: 51229.1641 -
I_var: 0.0158 - val_loss: 51617.8008 - val_I_var: -0.0082
Epoch 12/1000
91/91 [=====] - 1s 7ms/step - loss: 51209.5391 -
I_var: 0.0163 - val_loss: 51738.4023 - val_I_var: -0.0117
Training time: 11.5 seconds
Model saved to these files:
    splimlib_mpa_pairwise.pickle
    splimlib_mpa_pairwise.h5

```

```

Model loaded from these files:
    splimlib_mpa_pairwise.pickle
    splimlib_mpa_pairwise.h5

```

Allowing pairwise epistasis, the measurement process makes no sense: variants abundant in the low gate have middle phenotype? If anything, the results should be better or similar to the neighbour model.

Black box model

The black box model is a neural network model without a clear mapping to the shape of the G-P map, allowing more nonlinearity in the MAVE-NN function. We include it here to test the upper limit of the extent to which the package can learn the G-P pattern from our data.

N = 56,591 observations set as training data.
Using 20.1% for validation.
Data shuffled.
Time to set data: 0.271 sec.

Epoch 1/1000
91/91 [=====] - 0s 3ms/step - loss: 52004.3984 -
I_var: -0.0068 - val_loss: 51460.7852 - val_I_var: 6.3264e-04

Epoch 2/1000
91/91 [=====] - 0s 2ms/step - loss: 51797.9453 -
I_var: -0.0010 - val_loss: 51437.5547 - val_I_var: 0.0012

Epoch 3/1000
91/91 [=====] - 0s 2ms/step - loss: 51780.1992 -
I_var: -1.3591e-04 - val_loss: 51430.1719 - val_I_var: 0.0015

Epoch 4/1000
91/91 [=====] - 0s 2ms/step - loss: 51772.5391 -
I_var: -5.4145e-04 - val_loss: 51456.5391 - val_I_var: 7.2639e-04

Epoch 5/1000
91/91 [=====] - 0s 2ms/step - loss: 51780.3867 -
I_var: -1.6829e-04 - val_loss: 51421.4922 - val_I_var: 0.0017

Epoch 6/1000
91/91 [=====] - 0s 2ms/step - loss: 51749.1445 -
I_var: 4.8490e-04 - val_loss: 51443.9453 - val_I_var: 0.0010

Epoch 7/1000
91/91 [=====] - 0s 2ms/step - loss: 51746.1914 -
I_var: 3.1626e-04 - val_loss: 51468.1094 - val_I_var: 3.1849e-04

Epoch 8/1000
91/91 [=====] - 0s 2ms/step - loss: 51737.8945 -
I_var: 8.5977e-04 - val_loss: 51502.7578 - val_I_var: -7.2915e-04

Epoch 9/1000
91/91 [=====] - 0s 2ms/step - loss: 51720.3242 -
I_var: 9.7383e-04 - val_loss: 51504.4844 - val_I_var: -6.7569e-04

Epoch 10/1000
91/91 [=====] - 0s 2ms/step - loss: 51721.8672 -
I_var: 8.4948e-04 - val_loss: 51497.5469 - val_I_var: -5.0819e-04

Epoch 11/1000
91/91 [=====] - 0s 2ms/step - loss: 51699.9258 -
I_var: 0.0014 - val_loss: 51468.1719 - val_I_var: 3.1870e-04

Epoch 12/1000
91/91 [=====] - 0s 2ms/step - loss: 51697.2344 -
I_var: 0.0020 - val_loss: 51442.6445 - val_I_var: 0.0010

Epoch 13/1000
91/91 [=====] - 0s 2ms/step - loss: 51677.6172 -
I_var: 0.0021 - val_loss: 51488.9805 - val_I_var: -2.9304e-04

Epoch 14/1000
91/91 [=====] - 0s 2ms/step - loss: 51690.9844 -
I_var: 0.0023 - val_loss: 51490.4883 - val_I_var: -3.4275e-04

Epoch 15/1000
91/91 [=====] - 0s 2ms/step - loss: 51668.6602 -
I_var: 0.0026 - val_loss: 51565.4648 - val_I_var: -0.0024

Training time: 3.3 seconds
Model saved to these files:
 splimlib_mpa_blackbox.pickle
 splimlib_mpa_blackbox.h5

Model loaded from these files:
splimlib_mpa_blackbox.pickle
splimlib_mpa_blackbox.h5

Here, the measurement process looks similar to the first two (additive and neighbour), though not obviously better. Whatever the blackbox model learned, somehow it cannot be visualised with the usual G-P map.

Concluding comments:

1. The FACS-NGS data has more noise than the example included in the package publication; in fact, those data (sort-seq) had 10 bins, but only 0 or 1 values in the sorting matrix - as if each variant is strictly only sorted in a single FACS gate. We do not know in detail which data was used in Tareen et al., but their data look as if every sequence is only appearing in any one bin. In typical experimental FACS data, this will not be the case, as there is always some experimental variation that makes a sequence appear in more than one bin. In our dataset we have an experimental spread across the three bins (judging from the controls); with just three bins the resolution in the medium activity variants is low and oversampling also amplifies the spillover counts. While we are confident, based on the flow cytometry controls, that >92% of the sequences recovered from the high activity gate encode active variants (8% false *positive* rate), the high false *negative* rate (~50%) means that one out of two beads encoding an active variant are sorted in the low activity gate. Although this does not hamper the informative power of variants recovered from the high activity gate, and allows for pooled analysis of active variants, it does increase the noise, as the variants are not strictly sorted in one FACS gate. In other words, in this particular dataset a variant's enrichment in the high gate is informative, while the presence of some sequencing reads in the low gate does not provide useful information. Finally, the noisier the FACS-NGS data, the harder it is to rank the variants by phenotype (=generate a latent phenotype value).
2. When the MAVE-NN algorithms attempt to create the genotype-phenotype map, a uniform alphabet for all positions is assumed. However, our library is focused. Since all 13 possible values (12 amino acids or no amino acid) appear in all 6 positions, we suspect that the algorithm is trying to guess the preference for missing amino acids at all positions. Comparing this to the actual enrichment maps we show in the main paper, the algorithm is not actually succeeding in this task.

A combination of the above points takes the analysis back full circle: since it is hard to estimate the exact phenotype from the FACS-NGS data, it is not unreasonable to look at the active variants as a group. The targeted design of the library enables us to explore more positions and interactions in the D-domain given a limited sequencing depth, at the cost of requiring a targeted analysis of the results. We conclude that our exploration of enrichment patterns does show a fairly straightforward and biologically plausible picture despite possible limitations arising from noise.

Reviewers' Comments:

Reviewer #1:

Remarks to the Author:

The authors have adequately addressed my initial comments and concerns.

Reviewer #2:

Remarks to the Author:

First I want to thank the authors in addressing the questions raised by the three reviewers including myself. I can tell that the authors put much effort in composing the rebuttal letter, which is among most clearly written I have seen. Most of my questions have been satisfied. My contention is mostly focused on the authors analysis of their data using the software MAVE-NN that I recommended. I appreciate the authors' effort in trying to use MAVE-NN to analyze their data to estimate the additive and pairwise interaction terms. The authors provided a report of the MAVE-NN output and concluded that its model fits are not sensible nor consistent with their previous finding. However, I encourage the authors to rerun MAVE-NN and examine more of its output more carefully. I want to emphasize that I am not the author of MAVE-NN and this is not mandatory and the authors are free to opt out. But I do believe that using a principled method that takes into account the measurement noise to analyze the data will make the results a lot more robust and help resolve concerns about the current log enrichment ratio calculations raised by me and reviewer #3, and potentially other experts in the field.

For this round of revision, I encourage the authors (a). reexamine their previous model fit more thoroughly to check if MAVE-NN had indeed failed; (b). retrain the models with new hyperparameters if necessary; (c). report the new results; (d). replace the current enrichment ratios with the MAVE-NN inferred single and double mutant effects, if these results are good. I elaborate my suggestions below.

(1) a good way to measure the performance of MAVE-NN is to plot the distribution across bins for all possible latent phenotypic value. The authors did make this plot for all three models they fitted. However, I note that the plot ranges in the $p(y | \phi)$ vs. latent phenotype figures for the three models are all -5 to +3. I think it is very unlikely that the three models all happened to have the exact same range for their latent phenotypes. And since this plot range is the same as this example (https://github.com/jbkinney/mavenn/blob/master/mavenn/examples/tutorials/sortseq_mpa_visualization.png), I think it is probable that the authors used the same arbitrary plot range for all three methods. The actual dynamic range for the latent phenotype could be much wider than what is shown. This can explain why the distribution among bins are so uniform across the plotted range, since the plots only show the middle region of the phenotypic distribution. So I suggest the authors remake these figures using the actual range of ϕ .

(2) I think the authors have not provided enough output to examine the effect of single and double mutants. For all three models, the authors only provided sequence logos for the single mutant effects. While this is one way to visualize single mutational effects, the sequence logo is not very readable. And this makes it hard to compare the MAVE-NN results with the results in the paper which is presented in a position weight matrix. More importantly, for the two epistatic models, there should be a heat map showing the pairwise interaction strengths, similar to figure 4 in the manuscript. I believe there are helper methods in MAVE-NN that can automatically generate the heat map for both the single mutant and double mutant effects. Therefore, I suggest the authors make these figures and compare them with their existing results, for example by making a scatter plot and examine the R^2 .

(3) based on the model outputs (for example, in the pairwise model, the non-monotonic relation between bin distribution and phenotype, as well as the preference for deletions), I think there are reasons to believe that MAVE-NN might have not converged (in particular for the pairwise model) and/or the hyperparameters used were not optimal for this dataset. Therefore, I suggest the authors try different values for the regularization parameters and learning rate and examine if the

model had converged by plotting the validation loss against epochs and calculating the performance metric (predictive information or log likelihood) on the test set. All of these can be done following standard machine learning practice. Since MAVE-NN does not contain as many parameters as typical neural networks, finding the optimal hyperparameters should be relatively easy. And one way to tell if the models have worked is to see if the performance of the pairwise models are better than the additive model.

(4) the authors raised concerns that the low number of bin numbers might affect MAVE-NN's performance. I want to assure the authors that this is not a problem and that MAVE-NN's framework can work with any bin number larger than 1.

Reviewer #3:

Remarks to the Author:

The revised manuscript and supplement contain mainly new analyses, as well as new data and corrections suggested by referees, which altogether clarify and strengthen their work. In their revised version and rebuttal letter, the authors have addressed all of my comments. In short, their conclusions are robust with respect to the precise choice of the enrichment computation method, likely because the signal to noise ratio is very strong in their data (which is a striking feature of their new experimental setup). In particular, the library construction method seems to alleviate the need for initial frequency measurements as the latter seem to be very close to "ideal frequencies", which is a clearly strong asset of this method. I agree with reviewer 2 that deeper analysis of their dataset (e.g. statistical modeling) would be possible/informative and will certainly be performed by computational biologists on the published dataset in the near future. The current work describes a new and powerful experimental setup, and a first significant step in analyzing the experimental data, with robust conclusions regarding how signaling function is encoded in the sequence.

In my opinion their work is thus now ready for publication.

Responses to Reviewer #2

• *My contention is mostly focused on the authors analysis of their data using the software MAVE-NN that I recommended. I appreciate the authors' effort in trying to use MAVE-NN to analyze their data to estimate the additive and pairwise interaction terms. The authors provided a report of the MAVE-NN output and concluded that its model fits are not sensible nor consistent with their previous finding. However, I encourage the authors to rerun MAVE-NN and examine more of its output more carefully. (...) I do believe that using a principled method that takes into account the measurement noise to analyze the data will make the results a lot more robust and help resolve concerns about the current log enrichment ratio calculations raised by me and reviewer #3, and potentially other experts in the field.*

For this round of revision, I encourage the authors (a). reexamine their previous model fit more thoroughly to check if MAVE-NN had indeed failed; (b). retrain the models with new hyperparameters if necessary; (c). report the new results; (d). replace the current enrichment ratios with the MAVE-NN inferred single and double mutant effects, if these results are good. I elaborate my suggestions below.

We share the reviewer's enthusiasm for MAVE-NN and are grateful for the detailed suggestions (addressed below). The reviewer's very detailed guidance to using the software package was much appreciated, but - despite our best efforts - we have not been able to derive a meaningful additional data output that would have enabled reanalysis of the data.

(1) a good way to measure the performance of MAVE-NN is to plot the distribution across bins for all possible latent phenotypic value. The authors did make this plot for all three models they fitted. However, I note that the plot ranges in the $p(y | \phi)$ vs. latent phenotype figures for the three models are all -5 to +3. I think it is very unlikely that the three models all happened to have the exact same range for their latent phenotypes. And since this plot range is the same as this example (https://github.com/jbkinney/mavenn/blob/master/mavenn/examples/tutorials/sortseq_mpa_visualization.png), I think it is probable that the authors used the same arbitrary plot range for all three methods. The actual dynamic range for the latent phenotype could be much wider than what is shown. This can explain why the distribution among bins are so uniform across the plotted range, since the plots only show the middle region of the phenotypic distribution. So I suggest the authors remake these figures using the actual range of ϕ .

The reviewer is correct that our first attempt at reanalysis was closely modelled on the published SortSeq example in the package documentation and thus accidentally kept the default Phi range in the plots. After using the Phi values extracted from each model, the relevant range is closer to (-4, +4), which is kept in the new plots (**Rev. Fig. 1**).

Rev. Fig. 1: The measurement process in the MAVE-NN models previously submitted to reviewers, replotted to appropriate Φ range. The general evaluation of model performance remains unchanged; although the pairwise model is expected to perform better than additive or pairwise models, the measurement process shows an unexpected two-sided distribution.

(2) I think the authors have not provided enough output to examine the effect of single and double mutants. For all three models, the authors only provided sequence logos for the single mutant effects. While this is one way to visualize single mutational effects, the sequence logo is not very readable. And this makes it hard to compare the MAVE-NN results with the results in the paper which is presented in a position weight matrix. More importantly, for the two epistatic models, there should be a heat map showing the pairwise interaction strengths, similar to figure 4 in the manuscript. I believe there are helper methods in MAVE-NN that can automatically generate the heat map for both the single mutant and double mutant effects. Therefore, I suggest the authors make these figures and compare them with their existing results, for example by making a scatter plot and examine the R^2 .

We have added the single and pairwise interaction matrices, which are very different from each other depending on the model and appear rather random (see **Rev. Fig. 2**). Convinced that these particular models were not very good, we moved on to address the comments in (3) to examine the neural network training parameters and see if we could obtain better models.

Rev. Fig. 2: The modelled effect of single position mutations in three models. Red indicates a deleterious change and blue a beneficial mutation. Note the scales in the three plots are different.

(3) based on the model outputs (for example, in the pairwise model, the non-monotonic relation between bin distribution and phenotype, as well as the preference for deletions), I think there are reasons to believe that MAVE-NN might have not converged (in particular for the pairwise model) and/or the hyperparameters used were not optimal for this dataset. Therefore, I suggest the authors try different values for the regularization parameters and learning rate and

examine if the model had converged by plotting the validation loss against epochs and calculating the performance metric (predictive information or log likelihood) on the test set. All of these can be done following standard machine learning practice. Since MAVE-NN does not contain as many parameters as typical neural networks, finding the optimal hyperparameters should be relatively easy. And one way to tell if the models have worked is to see if the performance of the pairwise models are better than the additive model.

During re-training, we explored the following parameters:

- the number of nodes in the neural network, typically between 20 and 200,
- the learning rate: as expected, a lower value for the learning rate results in more epochs and more processing time needed for training, but the results are comparable (and non-informative) for values between 0.005 and 0.01,
- the number of epochs: tested between 50 and 200, found not to be critical to model training as models generally converged between epoch 20 and 40,
- the batch size during training: batch size refers to the number of training set data points used in each ‘sub-cycle’ of network training. We tested this parameter between batch size 10 and 1000, which generated different models with same very limited information encoded,
- using linear vs non-linear initialisation: this gave equally poor final performance.

Below we demonstrate example charts showing the outcome of one representative training run training pairwise models with different parameters. Compared to the default MAVE-NN settings, this run used 4 different network sizes (50 to 200 nodes), a slower learning rate, more epochs and a larger batch size (**Rev. Fig. 3**).

Additionally, the entire set of model training attempts was repeated with a reduced data set, which only contained:

- a) very active variants, which we know exhibit the Leu/Ile enrichment and epistasis patterns demonstrated in the main paper, and
- b) very inactive variants, which show the opposite pattern (depleted for Leu/Ile residues and the Φ -X- Φ motifs).

We re-ran all the MAVE-NN analyses described above on this dataset, hoping that the models would at least be able to reproduce the effects on single position mutant variants. Additionally, including fewer middle activity variants with noisy sequencing counts could facilitate neural network training. Unfortunately, the results were just as inconclusive.

Rev. Fig. 3: Training pairwise MAVE-NN models with different number of nodes results in poor models. In all cases, improvement on the training set is accompanied by loss of performance on the validation set.

Top: the heatmaps across four models show different results for each randomised position, depending on the arbitrary choice of network size. *Middle:* as the models learn, the amount of information learned on the training set (blue) climbs towards 0 during model learning, but performance on the validation set (orange) become worse with increasing epoch number. With information learned this close to 0, the models are effectively making random guesses. *Bottom:* After initial drop in the loss function on the training set (green), the model has converged to a constant value. However, performance on the validation set is in fact slightly worse than with a randomly initialised model. This further supports the conclusion that the models are not encoding true genotype-phenotype maps.

Parameters: learning_rate=0.001, epochs=200, batch_size=1000, stop=False, lin_init=True.

(4) the authors raised concerns that the low number of bin numbers might affect MAVE-NN's performance. I want to assure the authors that this is not a problem and that MAVE-NN's framework can work with any bin number larger than 1.

That is reassuring to know, and we will bear it in mind when we use MAVE-NN next.